# The natural abundance of stable water isotopes method may overestimate deep-layer soil water use by trees

Shaofei Wang[a], Xiaodong Gao[b,c,*], Min Yang[a], Gaopeng Huo[d], Xiaolin Song[e], Kadambot H. M. Siddique[f], Pute Wu[b,c], Xining Zhao[b,c]

[a]College of Water Resources and Architectural Engineering, Northwest A&F University, 712100, Yangling, Shaanxi Province, China
[b]Institute of Soil and Water Conservation, Northwest A&F University, 712100, Yangling, Shaanxi Province, China
[c]Institute of Soil and Water Conservation, Chinese Academy of Sciences and Ministry of Water Resources, 712100, Yangling, Shaanxi Province, China
[d]College of Land and Resources, Hebei Agricultural University, 071001, Baoding, Hebei Province, China
[e]State Key Laboratory of Crop Stress Biology for Arid Areas, College of Horticulture, Northwest A&F University, 712100, Yangling, Shaanxi Province, China
[f]The UWA Institute of Agriculture, The University of Western Australia, Perth, WA 6001, Australia

*Correspondence to*: Xiaodong Gao (gao_xiaodong@nwafu.edu.cn)

**Abstract.** Stable water isotopes have been used extensively to study the water use strategy of plants in various ecosystems. In deep vadose zone (DVZ) regions, the rooting depth of trees can reach several meters to tens of meters. However, the existence of roots in deep soils does not necessarily mean the occurrence of root water uptake, which usually occurs at a particular time during the growing season. Therefore, quantifying the contribution of deep-layer soil water (DLSW) in DVZ regions using the natural abundance of stable water isotopes may not be accurate because this method assumes that trees always extract shallow- and deep-layer soil water. We propose a multi-step method for addressing this issue. First, isotopic labeling in deep layers identifies whether trees absorb DLSW and determines the soil layer depths from which trees derive their water source. Next, calculating water sources based on the natural abundance of stable isotopes to quantify the water use strategy of trees. We also compared the results with the natural abundance of stable water isotopes method. The 11- and 17-year-old apple trees were taken as examples for analyses on China's Loess Plateau. Isotopic labeling showed that the water uptake depth of 11-year-old apple trees reached 300 cm in the blossom and young fruit (BYF) stage and only 100 cm in the fruit swelling (FSW) stage, whereas 17-year-old trees always consumed water from the 0–320 cm soil layer. Overall, apple trees absorbed the most water from deep soils (>140 cm) during the BYF stage, and 17-year-old trees consumed more water in these layers than 11-year-old trees throughout the growing season. In addition, the natural abundance of stable water isotopes method overestimated the contribution of DLSW, especially in the 320–500 cm soil layer. Our findings highlight that determining the occurrence of root water uptake in deep soils helps quantify the water use strategy of trees in DVZ regions.

## 1 Introduction

In the past three decades, water availability for vegetation growth has declined (Jiao et al., 2021), and the response of plants
to their water environment has received increasing attention (Eggemeyer et al., 2009; Nehemy et al., 2021; Wu et al., 2021).
Drought intensity and frequency are expected to increase with climate change (Huang et al., 2017; McDowell et al., 2016),
which will likely affect soil water availability and exacerbate changes in vegetation dynamics (Potts et al., 2006). In deep
vadose zone (DVZ) regions, trees generally develop deep roots that can access deep-layer soil water (DLSW), facilitating
transpiration, potentially buffering against drought stress and contributing to C sequestration in deep soils (Ding et al., 2021;
Fan et al., 2017; Germon et al., 2020; Nardini et al., 2016; O'Connor et al., 2021; Wang et al., 2022). Although the important
role of DLSW is well-established, few studies have quantified the contribution of DLSW to plant transpiration, limiting our
insight into how plants adapt to volatile water environments.

Analytical techniques based on stable isotopes ($\delta$D and $\delta^{18}$O) can be applied to study plant water use based on the
assumption that no isotopic fractionation occurs during root water uptake (Dawson et al., 2002; Ehleringer and Dawson,
1992; Evaristo et al., 2015; Rothfuss and Javaux, 2017). The isotopic comparison of xylem water and various water bodies
(e.g., soil water from different depths, underground water) could reveal the main water sources of plants when significant
differences in $\delta$D and $\delta^{18}$O occur between different water bodies (Ding et al., 2021; Yang et al., 2015; Zhao and Wang, 2021).
Although some recent studies found isotopic offset along the soil–root–stem–twig–leaf pathway (e.g., Barbeta et al., 2019;
Poca et al., 2019; Vargas et al., 2017), the mechanism of the fractionation remain in debate (Barbeta et al., 2022; Chen et al.,
2020; Wen et al., 2022; Zhao et al., 2016). Orlowski et al. (2016a,b, 2018) suggested that the fractionation is mainly related
to cryogenic vacuum distillation (CVD). However, the CVD is still the most common methodology for water extraction to
date (De La Casa et al., 2022). Plant water use strategies in various ecosystems have been researched extensively using
natural stable water isotopic techniques (Beyer et al., 2016; Dawson and Ehleringer, 1991; Jiang et al., 2020; Miguez-Macho
and Fan, 2021). However, it is challenging to quantify where in the soil profile the roots extract water due to limitations in
monitoring technologies and unclear physical processes such as preferential flow (Xiang et al., 2019; Zarebanadkouki et al.,
2013). Furthermore, most studies using the natural abundance of stable isotopes usually assumed soil water utilization at
specific depths based on vertical root distribution (Huo et al., 2018; Tao et al., 2021b; Wu et al., 2022; Zhao et al., 2020),
which may incorrectly quantify the contribution of different water sources, especially for DLSW.

Several recent studies have demonstrated that the presence of roots in the soil profile does not necessarily reflect where
plants are absorbing water (Ehleringer and Dawson, 1992; Kulmatiski et al., 2010; Szutu and Papuga, 2019), particularly in
deep soils. For example, Wang et al. (2020) argued that the absorption of deep soil water only occurred during the blossom
and young fruit (BYF) stage in apple orchards, whereas it rarely occurred in other stages. Therefore, understanding plant
water use strategies should priorly determine the soil layer depths from which trees derive their water source. Surprisingly,

few studies have determined where in the soil profile plants absorb water and how it varies during the growing season to quantify plant water use accurately. The isotopic labeling method could increase the isotopic abundance of deep soil water, providing direct evidence of plants' root water uptake (Beyer et al., 2018). Huo et al. (2020) investigated the water use strategy of agroforestry systems using natural isotopic signatures and isotopic labeling; however, the soil layer depths from which trees derived water (0–120 cm) was based on the distribution of intercrop roots rather than tree root water uptake. Isotopic labeling detected a pronounced uptake of soil water at 200 cm depth, suggesting an underestimation of the contribution of deep soil water to trees. Therefore, it might offer more reliable insights into the water use strategy of trees in DVZ regions by first identifying the soil layer depths from which trees derive their water source using isotopic labeling in deep layers and then calculating water sources based on the natural abundance of stable isotopes. In addition, root distribution and water use strategy often vary depending on stand age (Li et al., 2019; Wang et al., 2021a). More detailed information on root water uptake is required to elucidate stand age and growing season effect on ecohydrological processes for sustainable vegetation development in DVZ regions.

China's Loess Plateau is a typical dryland ecosystem with DVZ, severe drought and water shortages occur (Fu et al., 2017). Most vegetation in this region grows under rainfed conditions and develops deep root systems (Wang et al., 2022; Wang et al., 2015; Yang et al., 2022). On the Plateau, apple trees are the dominant cash tree plantations. Over recent decades, the cultivated area of apple trees increased continuously, with the Plateau becoming the largest apple tree cultivation zone globally, accounting for more than one-quarter of global coverage and production (Gao et al., 2021b). The apple industry has become the backbone of the local rural economy, involving more than 10 million farmers (Gao et al., 2021b). However, severe drought and water shortages and intense seasonal precipitation variation have resulted in a complicated and volatile soil water environment in this region, hampering the sustainability of apple trees. Therefore, this study aimed to (a) identify the dynamics of DLSW absorption for apple trees, (b) ascertain the water use strategy response of apple trees to variations in the growing season and stand age, and (c) elucidate the difference between the combined method (combining isotopic labeling in deep soils with natural stable isotope signatures) and the natural abundance of stable water isotopes method.

## 2 Materials and Methods

### 2.1 Study area and experimental site

The study was conducted in 2019 in Chengcheng county, Shaanxi Province, China, in the temperate continental monsoon climate zone. Mean annual precipitation in the study region is 507.9 mm/y, and the annual average temperature is 12.6 ℃/y (1999–2018). A thick layer of loess covers the land surface, and the groundwater level is over 50 m depth on average, which cannot be absorbed by plant roots. Apple (*Malus pumila Mill.*) orchards of two stand ages (11- and 17-year-old) were selected (Fig. 1) to collect soil and xylem samples. The orchards are located in the same small watershed, with similar slopes, aspects, and soil texture, and subjected to the same management regimes (e.g., plant and row spacing (4 × 4 m), no irrigation,

standard clipping and fertilization). The sampling locations, height, trunk diameter at 80 cm height, and crown dimensions (long and short axes) of trees were recorded. General information on the apple orchards is in Tables 1 and S1.

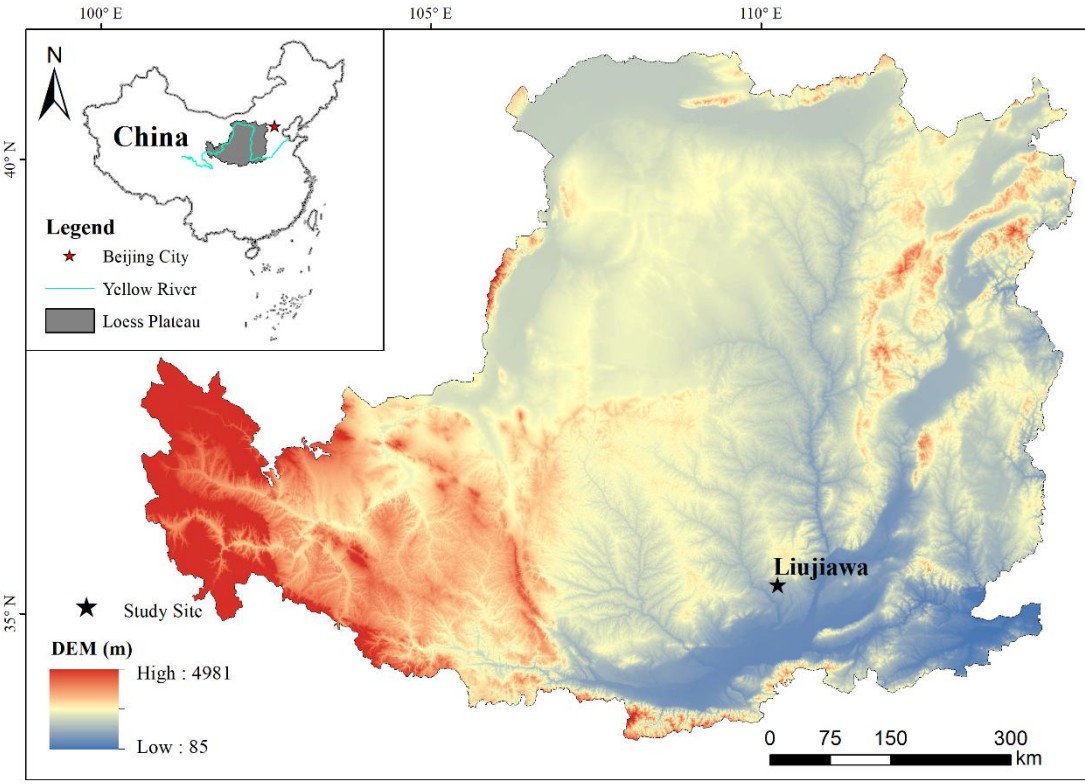

Figure 1: Location of sampling site on the Loess Plateau of China

Table 1. General information on the two apple orchards.

| Stand age (a) | Longitude | Latitude | Altitude (m) | Height* (cm) | Trunk diameter*(cm) | Crown* size (cm) |
|---|---|---|---|---|---|---|
| 11 | 109 °50'13" | 35 °20'5" | 863.8 | 355 | 12.0 | 405×352 |
| 17 | 109 °50'18" | 35 °19'58" | 862.9 | 395 | 14.4 | 450×380 |

*Height, trunk diameter and crown size were mean values for 20 trees in each orchard. Trunk diameter measured at 80 cm height.

## 2.2 Sample collection

A method combining isotopic labeling and the natural abundance of stable isotopes was used to investigate the water use strategy of apple trees. Firstly, isotopic labeling in deep layers was used to identify whether trees absorb DLSW and to determine the soil layer depths from which trees derive their water source. Next, we used the natural abundance of stable isotopes method to quantify the water use strategy of trees. Plant and soil samples in two experiments were collected at three

developmental stages in 2019: blossom and young fruit (BYF, May), fruit swelling (FSW, July), and fruit maturation (FTM, September).

### 2.2.1. Isotopic labeling experiments

Twelve trees with similar growth in each orchard were randomly selected for labeling at different soil depths (1 m, 2 m, 3 m, 4 m) (Fig. S1). Day 1 before $D_2O$ injection (May 1, July 9, September 3, 2019), four holes were drilled in quartering
radiation from each trunk (0 °, 90 °, 180 °, 270 °) at 50 cm radial distance. A long polyvinyl chloride pipe was inserted into the holes at the target depth before injecting 300 mL tracer solution ($\delta D$ = 714,000‰, 30 mL 99.99% $D_2O$ plus 270 mL tap water) into each hole. The total amount of injected solution was 1,200 mL for each tree. Huo et al. (2020) demonstrated that the change in soil water content (SWC) caused by 300 mL water is less than 1% on the Loess Plateau, with a negligible impact on soil water balance. After tracer injection, the polyvinyl chloride pipe was removed, and the hole was sealed.
Xylem samples from labeling trees were collected on days 1, 3, 5, and 7 after injection, and corresponding samples were collected from unlabeled trees before injection to obtain background isotope concentrations. Two xylem samples were collected for each tree, with a total sample size of six for each treatment in a single sampling. The background value was an average of six xylem samples from unlabeled trees. If the D concentration of the xylem sample was at least two standard deviations (SD) higher than the background value, the tracer was assumed to be present (Kulmatiski et al., 2010).

### 2.2.2. Collection of soil and vegetation samples for isotopic analysis

At each sampling event, three trees in each orchard were selected randomly. For xylem samples, three suberized twigs (0.5–1 cm in diameter) were cut from the sunny side of trees, and the bark, phloem, and cambium were removed. Each twig was cut into 1 cm segments and immediately placed in a 15 mL glass vial. The vial was sealed with parafilm, and placed in a box containing ice packs to prevent evaporation. After xylem sampling, soil samples were collected with a hand auger from 0–
500 cm soil profile (at 10, 20, and 40 cm intervals from the 0–20, 20–160, and 160–500 cm layers, respectively). One part of each soil sample was placed in a 100 mL vial and stored as per the xylem samples for isotopic determination, while the other was placed in an aluminum box to determine gravimetric SWC by oven-drying.

### 2.2.3 Sampling of rain water

Rain water samples (N = 32) were collected using a combined device of polyethylene bottle and funnel during rainfall events
between May and September. A plastic ball was placed on the funnel to prevent evaporation. The collected rain water samples were immediately sealed into vials by parafilm and stored at 4 ℃ for isotopic analysis.

## 2.3 Root data collection

In the FTM stage, a hand auger with 60 mm internal diameter was used to collect root samples (50 cm from the tree trunk) and three trees in each orchard were selected randomly for sampling. The samples were collected down to 500 cm in 20 cm
increments. The samples with roots were washed carefully with tap water in a sieve (0.2 mm) to remove all the soil. The root samples obtained were scanned using a scanner at 300 dpi, and then the fine root length was determined using WinRhizo

software (version 5.0 Regent Instruments Inc., Quebec, Canada). The fine root length density (FRLD) in each sample was calculated by dividing the length of fine roots by the volume of the sample.

## 2.4 Isotopic analysis

A CVD system (Li-2000; LICA United Technology Limited, Beijing, China) was used to extract water under a heating temperature of 95 °C and a pressure of 0.2 Pa, which has been applied in previous studies (Huo et al., 2020; Tao et al., 2021a; Wang et al., 2021b; Zhao et al., 2020). The extraction time of soil water and xylem water samples were 90 min and 120 min respectively. Samples were weighed before and after extraction and again after oven-drying for 24 h to calculate the extraction efficiency (Wang et al., 2021b). Samples with an extraction efficiency less than 98% were discarded. The stable
hydrogen and oxygen isotope compositions of extracted soil water and xylem water were determined using a TIWA-45EP isotope ratio infrared spectroscopy analyzer (Los Gatos Research, Mountain View, USA) and Stable Isotope Ratio Mass Spectrometer (Isoprime Limited, UK), respectively. The measurement precision of $\delta^{18}O$ and $\delta D$ is 0.2‰ and 1.0‰ for the TIWA-45EP isotope ratio infrared spectroscopy analyzer and 0.3‰ and 2.0‰ for the Stable Isotope Ratio Mass Spectrometer, respectively. Each isotopic sample was repeatedly measured six times. The first three measurements were
discarded to mitigate the memory effect of isotopic measurement, and the mean value of the last three measurements was taken as the isotopic value of sample.

In general, soil water is the primary water source for trees on the Loess Plateau. We assessed the isotopic offset between xylem water and soil water using soil water line conditioned excess (SW-excess) proposed by Barbeta et al. (2019):

160                          $$SW\text{-}excess = \delta D - a_s\delta^{18}O - b_s \tag{1}$$

where $a_s$ and $b_s$ are the slope and intercept of soil water line (SWL), respectively; $\delta D$ and $\delta^{18}O$ are the isotopic compositions of xylem water. A positive SW-excess value means that xylem water plots above SWL in a $\delta D - \delta^{18}O$ diagram (i.e. D in xylem water is more enriched than SWL), while a negative value means that xylem water plots below SWL in a $\delta D - \delta^{18}O$ diagram (i.e. D in xylem water is more depleted than SWL).

## 165  2.5 Determination of plant water sources

The Bayesian isotope mixing model, MixSIAR (version 3.1.7) package in R was used to calculate the contributions of soil water from different layers to xylem water (Stock and Semmens, 2013). Based on the distributions of soil water isotopic values and the results of labeling experiments, the 0–320 cm soil profile was divided into four water sources (0–40 cm, 40–140 cm, 140–240 cm, and 240–320 cm layers). The SWC and isotopic values varied the most in the shallow soil layer (0–40
cm) but were most stable in the deep soil layer (140–320 cm). The $\delta D$ and $\delta^{18}O$ values for each potential water source were used as source data; the $\delta D$ after subtracting the SW-excess and $\delta^{18}O$ values for xylem water were used as mixture data for the model. $\delta D$ values of xylem water corrected by SWL can match those of soil water. Thus, the fractionation factor was set to zero, assuming no isotopic fractionation during root water uptake.

## 2.6 Statistics

A one-way analysis of variance was used to determine differences in SWC, $\delta$D values, and the contribution of water sources among sampling events. The least significant difference was used to perform the post-hoc analysis, with significance evaluated at the 0.05 level ($P < 0.05$). The statistical analyses were carried out in SPSS 23.0, with all figures drawn using Origin 2016.

## 3 Results

### 3.1 Precipitation and temperature distribution

Figure 2 shows the total precipitation ($P_t$) and growing season (April to September) precipitation ($P_g$). $P_t$ and $P_g$ in 2019 were 522.1 mm and 442.3 mm, respectively, similar to the multiyear (1999–2018) means (507.9 mm/y for $P_t$ and 407.5 mm/y for $P_g$). Seasonal variation was significant for 2019 precipitation, with 74.9% occurring from June to September, according with the seasonal distribution characteristics of precipitation in the Loess Plateau. In this way, the year of 2019 was considered a normal precipitation year. Additionally, the highest monthly precipitation and highest single precipitation event occurred in August and September, being 128.3 mm and 57.3 mm, respectively.

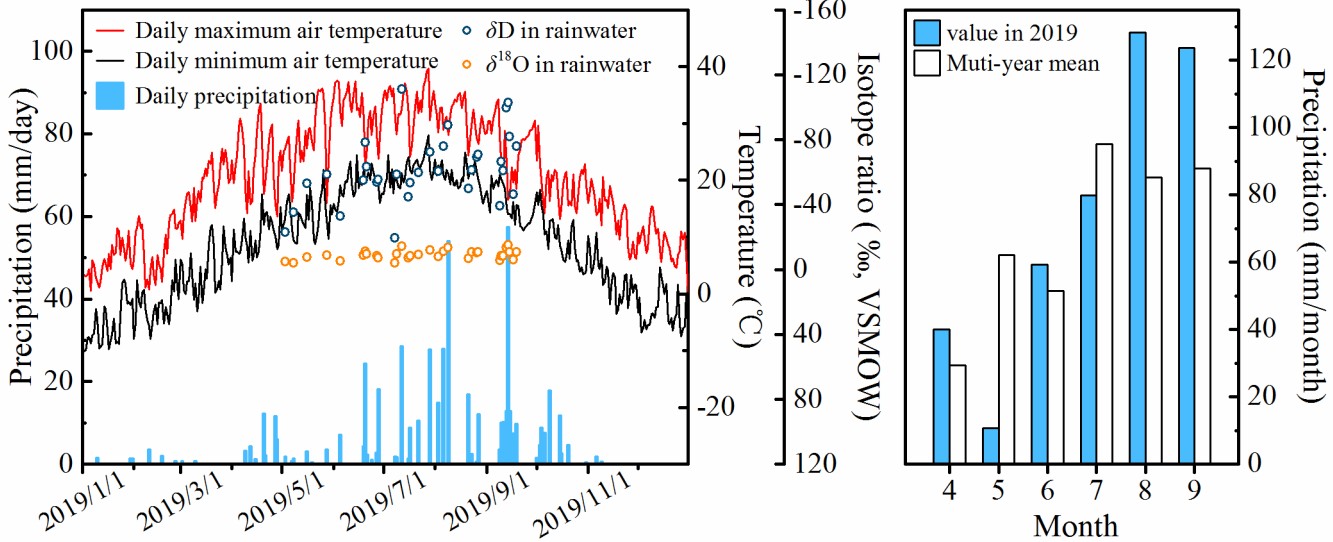

**Figure 2: Time series of meteorological data and rain water isotopic values in 2019, monthly precipitation in 2019, and multi-year mean, respectively.**

### 3.2 Root distribution and soil water content

Apple trees had dimorphic rooting systems, with fine roots distributed in shallow and deep soil layers (Fig. 3). The 11- and 17-year-old apple trees had similar root distribution profiles, with FRLD decreasing with increasing soil depth and the

maximum FRLD occurring at 40 cm and 60 cm depth, respectively. Overall, 17-year-old apple trees had more fine roots in the whole profile than 11-year-old trees. Similarly, the SWC values of the two apple orchards had similar temporal and spatial variations in the profile (Fig. 4). The SWC was highly variable in the 0–100 cm soil layer during the sampling period due to rainfall infiltration, soil evaporation, and root water uptake but was relatively stable in the 240–500 cm layer. The 17-year-old orchard had greater seasonal variations and lower mean SWC values than the 11-year-old orchard (Fig. S2).

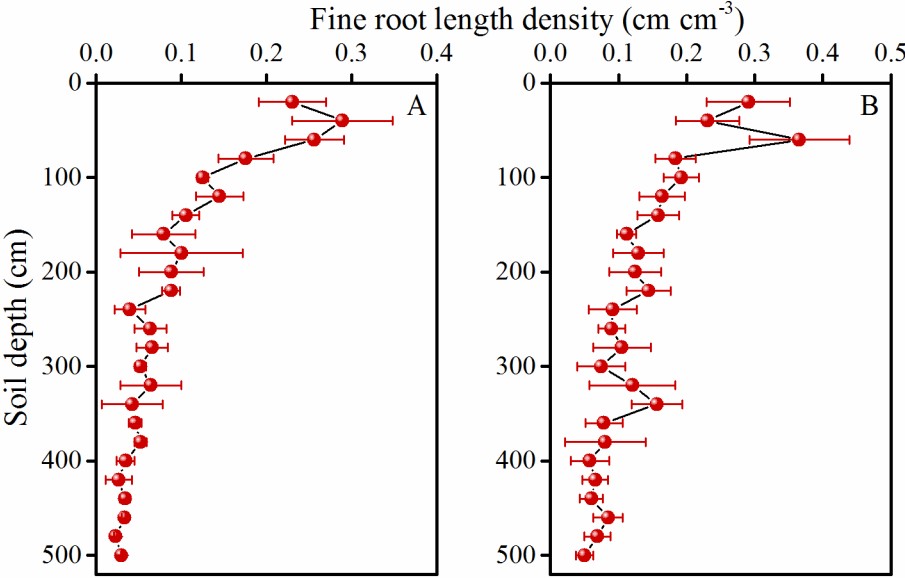

**Figure 3: Vertical distribution of fine root length density (FRLD) in 11-year-old (A) and 17-year-old (B) apple orchards. Values are means ±SD (N=3).**

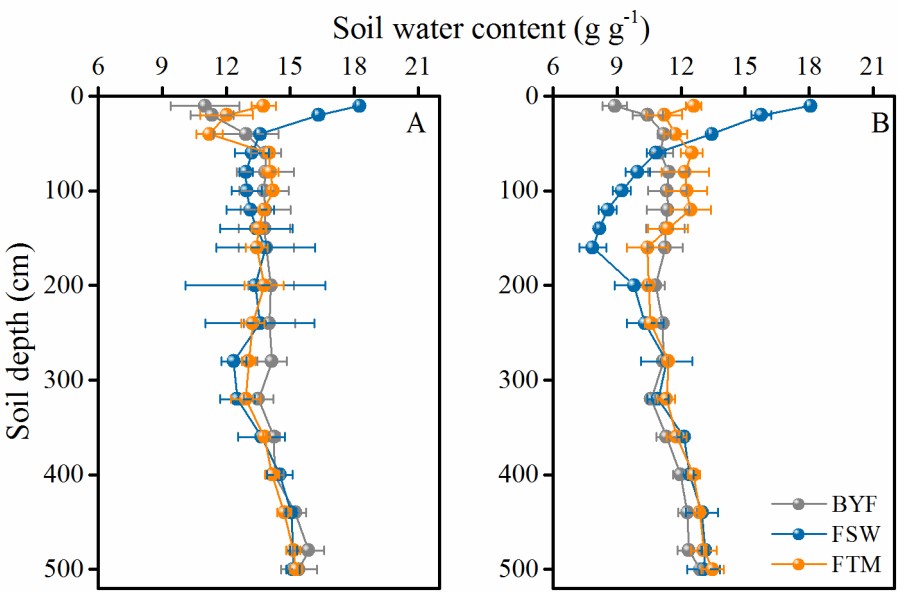

**Figure 4: Vertical distribution of soil water content (SWC) before the tracer injection in 11-year-old (A) and 17-year-old (B) apple orchards. Values are means ± SD (N=3).**

### 3.3 Absorption dynamics of $D_2O$ tracer

205     Root water uptake dynamics of apple trees are measured using the dynamics of $\delta D$ values in xylem water following labeling (Fig. 5). During the BYF stage, no tracer signal was detected at 4 m depth, while pronounced uptake of artificial tracer occurred at 1 m, 2 m, and 3 m depth for both apple orchards. During the FSW stage, tracer signals occurred in both apple orchards. Specifically, the 11- and 17-year-old apple trees quickly absorbed water from the 1 m soil layer, reflected in the markedly elevated $\delta D$ value for the xylem samples taken one day after $D_2O$ injections. The maximum concentration of D in

210     xylem for both apple orchards occurred on day 3 after labeling, then decreased rapidly, and was lower than background values on day 7. Moreover, the presence of artificial D was found in the 17-year-old orchard at 2 m and 3 m labeling depth, with the tracer peak occurring on day 3 after labeling, while none of the sampled trees in the 11-year-old orchard extracted water from soil profiles labeled at 2 m or deeper. During the FTM stage, both orchards had similar tracer uptake patterns to the FSW stage, with the peak $\delta D$ value in xylem water occurring on day 3 or 5 after labeling.

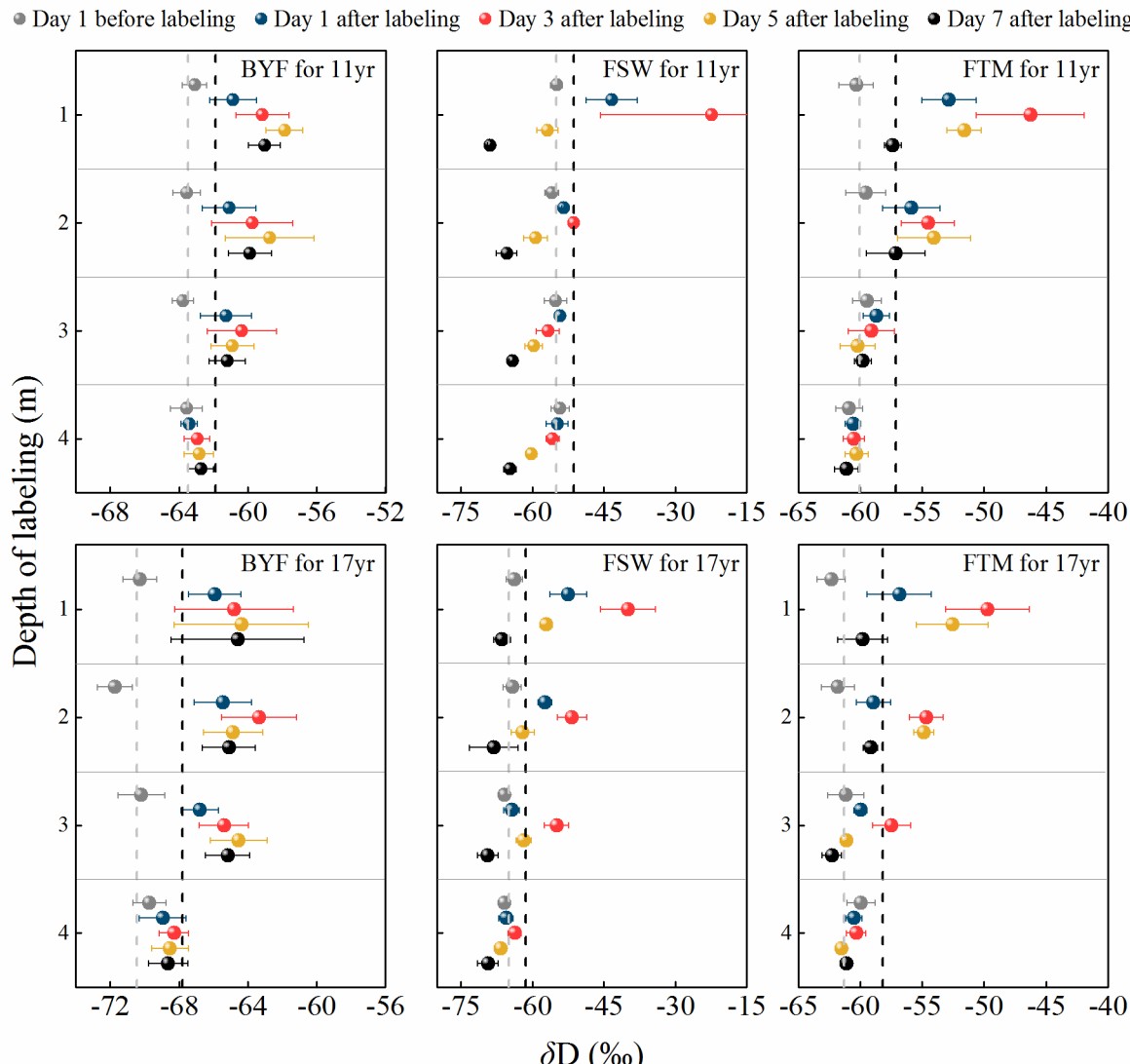

**Figure 5: Temporal dynamics of $\delta$D values in xylem water for 11- and 17-year-old apple trees. Sample collection started on day 1 before D$_2$O tracer solution ($\delta$D = 714,000‰) injection and commenced until day 7 (N=6). Gray dashed lines represent the background value (mean $\delta$D values in xylem water on day 1 before labeling) and black dashed lines represent 2 SD above the background value.**

## 3.4 Stable isotopes in xylem and soil water

The $\delta$D and $\delta^{18}$O values in soil water from both apple orchards had similar seasonal and vertical variations (Fig. 6). Shallower soils had more enriched isotopic values than deeper soils. The $\delta$D and $\delta^{18}$O values in the 0–40 cm soil layer varied significantly between sampling dates, attributed to the shallow infiltration of rainfall with negative isotopic values and intense surface evaporation. The isotopic values in the 100–500 cm soil layer did not significantly differ between sampling dates ($P > 0.05$).

Isotopic values in xylem water depended on growing season stage and stand age (Fig. S3). Specifically, the BYF stage had more depleted $\delta D$ and $\delta^{18}O$ values for 11- and 17-year-old apple trees than the FSW or FTM stage. The similar isotopic values for xylem water in apple trees may be due to the same or similar water sources, with different isotopic values signifying different water sources.

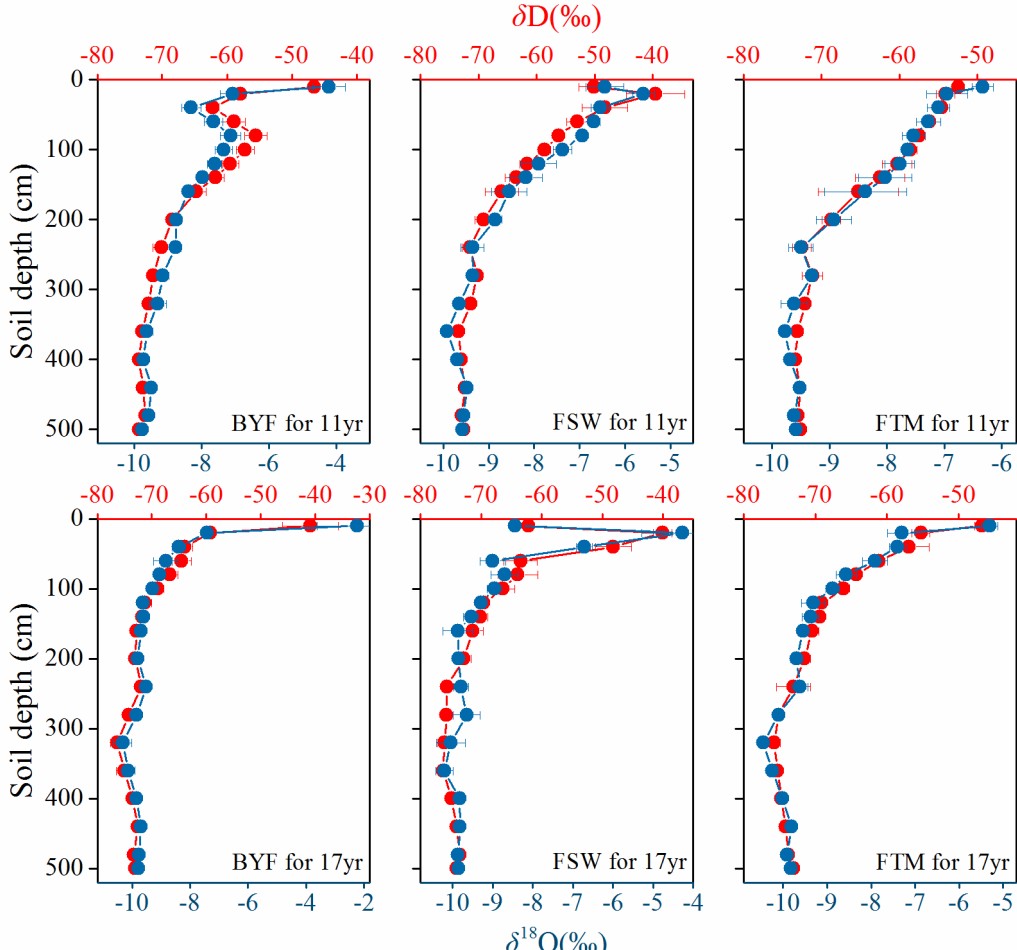

**Figure 6: $\delta D$ and $\delta^{18}O$ values in soil water down the soil profile during the apple growing season in 2019. Values are means $\pm$ SD (N=3).**

### 3.5 Differences between stand age and seasonal variations in water sources

The soil layer depths from which apple trees derived their water source during different growth stages were determined using isotopic labeling, before calculating the contribution proportion of different water sources to xylem water using the Bayesian mixing model. As shown in Fig. 7, isotopic values in xylem and soil water followed a similar trend to the local meteoric water line (LMWL). Across all samples, most of these isotopic values were plotted to the right of LMWL, indicating that

soil water in the study area came from precipitation and experienced intense evaporation. The relationship between isotopic values in xylem water and soil water revealed significant variation with growth stage and stand age, indicating that apple trees could extract water from different soil layers (Figs 7 and 8). Specifically, the BYF stage produced more negative isotopic values in xylem water for 11- and 17-year-old apple trees (Fig. 7), which mainly used water from the 140–320 cm soil layer (more than 48%) (Fig. 8). However, as precipitation infiltrated into subsurface layers, more positive isotopic values in xylem water occurred during the FSW and FTM stages. The isotopic values in xylem water were similar to it in the 0–40 cm soil layer (Fig. 7). Results from the mixing model revealed that the contribution of the 0–40 cm soil water reached 70% for 11-year-old apple trees during the FSW stage and 49% for 17-year-old apple trees during the FTM stage (Fig. 8). Overall, older apple trees relied more on deeper soil water during the growing season (Fig. 8).

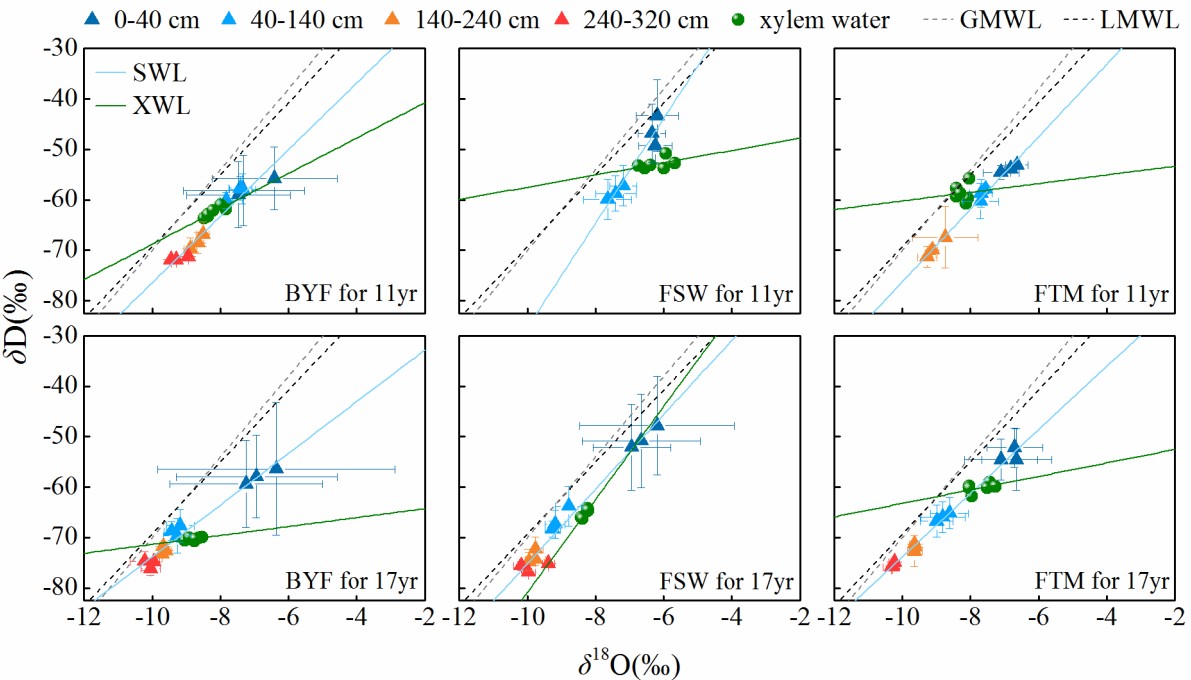

**Figure 7: $\delta$D and $\delta^{18}$O values in xylem water and different soil layers (0–40 cm, 40–140 cm, 140–240 cm, 240–320 cm) for 11- and 17-year-old apple trees (±SD). GMWL and LMWL represent global and local meteoric water lines, respectively. LMWL: $\delta$D = 7.1$\delta^{18}$O+2.1. SWL and XWL represent the soil and xylem water lines, respectively.**

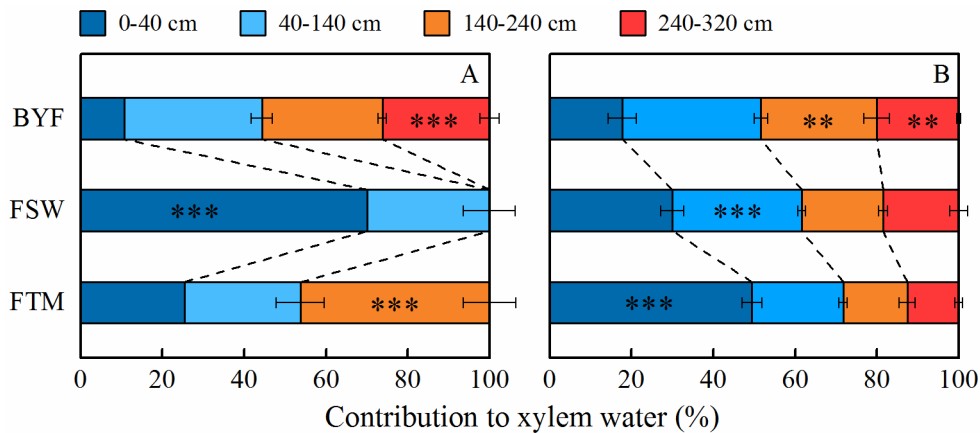

**Figure 8: The contribution of four potential water sources to xylem water in 11-year-old (A) and 17-year-old (B) apple trees. Error bars indicate standard errors of the means (N=3). Asterisks represent significant differences between growing stages (\*, $P < 0.05$; \*\*, $P < 0.01$; \*\*\*, $P < 0.001$).**

### 3.6 Relationship between the contribution of water sources and corresponding soil water content

The relationship between the contribution of water sources and SWC in apple orchards depended on soil depth (Fig. 9). When the data sets for a given soil layer were pooled, a significant ($P < 0.05$) and positive relationship occurred between SWC and its contribution in the 0–40 cm soil layer, indicating that apple trees increased the utilization proportion of deeper soil water as SWC decreased in the 0–40 cm soil layer. It is worth noting that the relationship also depended on stand age. Specifically, a significant relationship in the 0–40 cm soil layer occurred in the 11-year-old apple orchard but not in the 17-year-old apple orchard. In addition, after pooling the data sets for a given soil layer, there was a significant ($P < 0.05$) relationship between SWC and its contribution in the 140-320 cm soil layer.

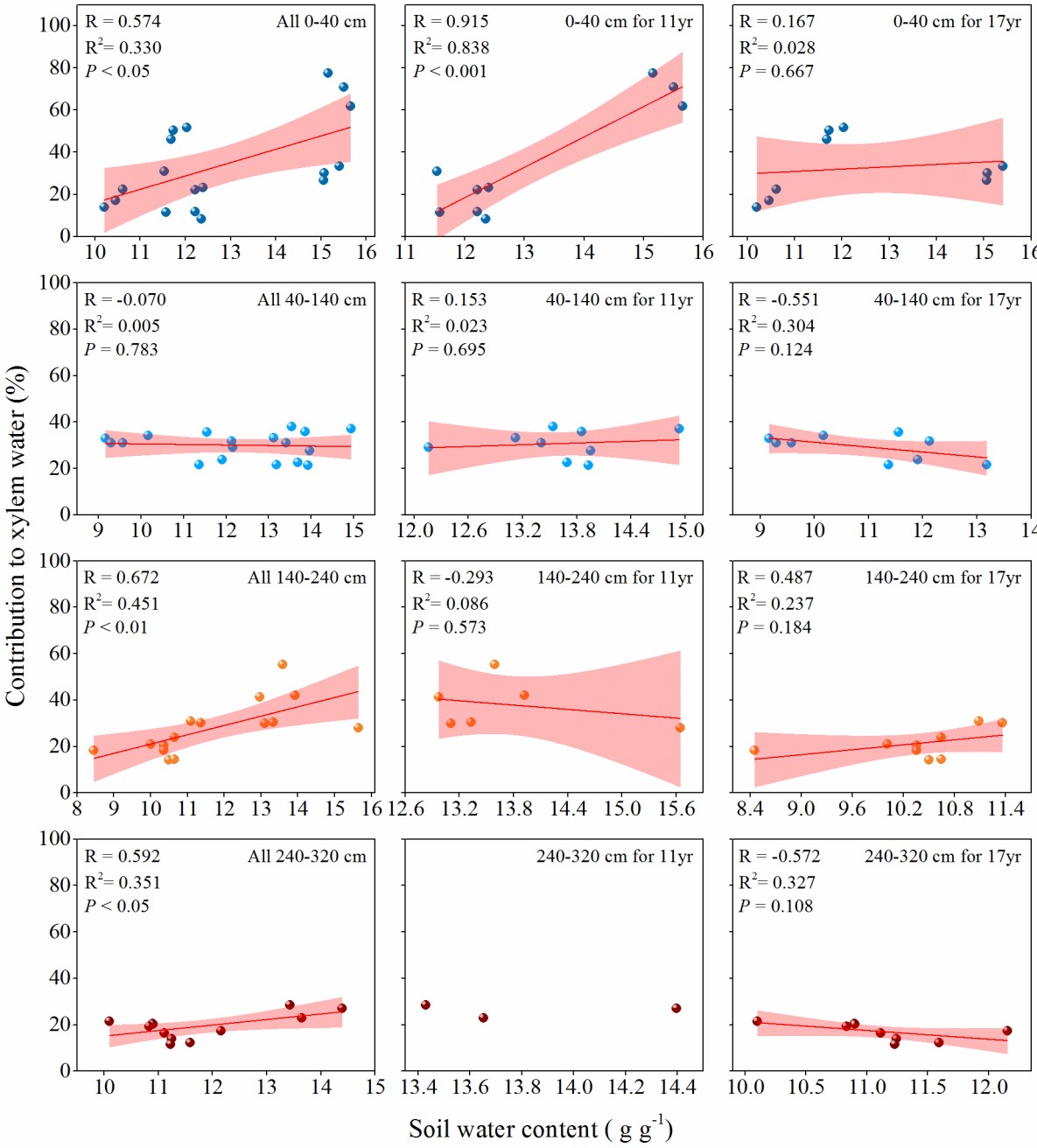

**Figure 9: Relationship between the contribution of water sources and soil water content in different soil layers of 11- and 17-year-old apple orchards.**

# 4 Discussion

## 4.1 Deep soil water uptake dynamics

Accurately determining the soil layer depths from which plants derive their water source is essential for quantifying their water use strategy in DVZ regions. However, it is often difficult to effectively capture changes in this depth caused by precipitation infiltration, affecting our evaluation of the contribution of different water sources to plant transpiration. In this study, we artificially injected $D_2O$ tracer to different soil layers to change soil water isotopic composition; then, root water uptake dynamics in the profile could be acquired by monitoring the concentrations of D in xylem samples, providing direct evidence for accurately determining root water uptake (Couvreur et al., 2020; Mennekes et al., 2021).

None of the apple trees in either orchard extracted water from the 4 m labeling depth during the BYF stage, but significant uptake of artificial tracer occurred at 1 m, 2 m, and 3 m depths (Fig. 5). This indicates that root water uptake in the 0–4 m soil layer is likely to meet the apple tree's transpiration, despite recent studies showed that the rooting depth of mature apple trees exceeded 5 m (Li et al., 2019; Wang et al., 2021c). We observed a distinctly different absorption of $D_2O$ tracer at different labeling depths in two orchards during the FSW stage (Fig. 5). At the 1 m labeling depth, an immediate reaction to $D_2O$ tracer was observed for the 11- and 17-year-old apple trees, in contrast to Evaristo et al. (2019) and Magh et al. (2020), who reported notable delays (days to weeks) between tracer injection and detection within xylem samples. This could be ascribed to the relatively tall canopy (>20 m) of their research trees, increasing the time taken for water to travel from roots to crown branches relative to apple trees. Unlike previous studies (Kahmen et al., 2021; Seeger and Weiler, 2021), the peak tracer concentration occurred on day 3 and then rapidly declined until day 7, possibly due to the heavy precipitation event with more depleted isotope on day 3. At the 2 m and 3 m labeling depths, $D_2O$ tracer was found in the 17-year-old orchard, while none of the 11-year-old sampled trees extracted water from soil profiles labeled at 2 m and below (Fig. 5). This finding indicates that the 17-year-old apple trees absorbed more water from deep soils than the 11-year-old trees, consistent with the findings of Wang et al. (2020). The FTM stage had similar labeling results to the FSW stage, where xylem water in 11-year-old trees had more enriched isotopes at 1 m labeling depth and more depleted isotopes at 2 m and below than 17-year-old trees.

## 4.2 Differences in water sources during the growing season and between stand ages

Numerous evidence from various ecosystems suggests that trees can adjust their water sources to adapt to changes in the surrounding water environment (Barbeta et al., 2015; Gao et al., 2018a; Ma et al., 2021; Zhao et al., 2021). In DVZ regions such as the Loess Plateau, soil water from precipitation is the primary water source for plant transpiration (Gao et al., 2021a; Tao et al., 2021a; Wu et al., 2022). Severe drought and water shortages and intense seasonal precipitation variation result in complicated and volatile soil water environments in this region. Hence, it is vital that apple trees adapt to this environment to survive and grow.

The contribution proportion of different soil water sources to plant transpiration was calculated using a Bayesian mixing model based on the depth of the tree's root water uptake determined using isotopic labeling. The isotopic signatures in xylem water and soil water depended on growth stage and stand age, such that water uptake patterns differed between apple tree growth stage and stand age (Fig. 7). The results from the Bayesian mixing model confirmed this finding (Fig. 8). More specifically, 11- and 17-year-old apple trees exhibited flexible water use strategies, shifting the main water source from deep to shallow soil layers as the growing season progressed (Fig. 8). This is consistent with several recent studies in this region (Huo et al., 2020; Zhao et al., 2020), in which root systems enabled trees to exhibit seasonal water use patterns, switching their water source between soil layers based on available soil water. Notably, SWC of the 0–40 cm soil layer significantly increased in apple orchards due to precipitation recharge during the FSW stage ($P < 0.01$) (Fig. 4), and 11-year-old trees rapidly changed their main water source to shallow soil layer (0–40 cm) (70%) in this stage, while 17-year-old trees changed their main water source to shallow soil layers (49%) until the FTM stage (Fig. 8). One possible explanation is that drought and high temperatures before sampling caused a less reversible embolism, reducing the water conductivity of the shallow root system (Grossiord et al., 2017). This difference indicates that the root water uptake response to soil water change depends on stand age, with 11-year-old trees more sensitive than 17-year-old trees in the 0–40 cm soil layer, and verified by the relationship between the contribution of water sources and SWC in the 0–40 cm soil layer (Fig. 9). Similarly, Huo et al. (2018) and Wang et al. (2021a) reported that old trees tended to access stable deep water sources and had a time lag converting water sources following a soil water change. Overall, the apple trees in both orchards absorbed the highest proportion of water in the 140–320 cm soil layer during the BYF stage. Moreover, 17-year-old apple trees had a higher proportion of water from these layers than 11-year-old trees throughout the growing season.

### 4.3 Implications

Plant root water uptake is sensitive to changes in the water environment, and changes in water uptake strategy affect their ecosystem functioning. In DVZ regions, accurately calculating the contribution proportion of different water sources to transpiration helps us understand the variation in plant water use strategies. However, few studies have determined the soil layer depths from which trees derive their water source when calculating the contribution of water sources based on the natural abundance of stable isotopes method. Numerous studies have determined this depth based on prior information (e.g., vertical root distribution) (Tao et al., 2021b; Wu et al., 2022; Zhao et al., 2021), which can reach 5 m to even 10 m. The existence of roots in deep soils does not necessarily mean that root water uptake occurs (Szutu and Papuga, 2019). The isotopic labeling results showed that no water uptake occurred in deep (>140 cm) soils for 11-year-old apple trees during the FSW stage (Fig. 5). Thus, quantifying the water use strategy of trees in DVZ regions using the natural abundance of stable water isotope method may not be accurate as this method assumes that trees always extract shallow- and DLSW. This study used isotopic labeling in deep layers to identify whether trees absorb DLSW and determine the soil layer depths from which trees derive their water source. Then we calculated water sources based on the natural abundance of stable isotopes method,

and compared the results with the natural abundance of stable water isotopes method (Figs 8 and S4). The results showed

that the natural abundance of stable water isotopes method overestimated the contribution proportion of DLSW (>140 cm) for 11-year-old apple trees (17%) during the FSW stages. The root water uptake depth from isotopic labeling indicated that soil water in the 0–140 cm layer replenished by precipitation could meet the transpiration demand of apple trees during the FSW stage (Figs 5 and 8), improving our understanding of the relationship between the contribution of water sources and SWC. The contribution proportion of water sources in the 0–40 cm soil layer increased significantly with SWC in the 11-

year-old apple orchard ($P < 0.001$), similar to the findings of previous studies (Gao et al., 2018b; Grossiord et al., 2017). In contrast, no significant differences occurred between the contribution proportions from the 2.4–3.2 m and 2.4–5 m soil layers in the 17-year-old apple orchard (Figs 8 and S4), and trees did not actually acquire water from the 3.2–5 m soil layer. This further indicates that determining the soil layer depths from which trees derive their water source is important for understanding the role of DLSW on plants, especially those with flexible water uptake strategies.


The apple trees on the Loess Plateau are heavily dependent on soil water in deep layers due to low annual precipitation (400-600 mm) and abundant soil water resource in DVZ (Wang et al., 2021c; Yang et al., 2022). Our results show that apple trees switch their water sources between different soil layers to adapt to the changing water environments on the Loess Plateau, which is particularly important in the context of climate change. Special attention should be directed to water consumption in

deep soils—we found that apple trees absorbed the most water from deep soils during the BYF stage, with 17-year-old apple trees consuming more water in these layers than 11-year-old trees throughout the growing season. This result is in accordance with previous observations in this region that soil water availability gradually decreased with increasing stand age, and then apple trees absorbed more water from deeper soil layers (Li et al., 2019). Similarly, Barbeta et al. (2015) found that trees increased their use proportion of deep soil water and groundwater following a long-term (12 years) experimental

drought. However, this water-use strategy may not be sustainable for trees in DVZ regions where deep soil water is difficult to be recharged. The result of the tritium peak method suggested that it took more than 50 years for soil water migration to 6 m depth in apple orchards in DVZ regions (Li et al., 2018). Thus, once DLSW is depleted, it cannot be replenished within a short timeframe, reducing the tree's ability to resist water stress. Also, Wu et al. (2021) observed that soil water generated by precipitation was the primary water source for apple trees when deep soil water was depleted, dominating their transpiration.

In this case, trees were likely to encounter irreversible embolism, increasing the risk of drought-induced mortality, threatening the sustainable development of vegetation and changing regional hydrological cycle (Brodribb et al., 2020; Zhang et al., 2020). Therefore, we suggest long-term and high-frequency monitoring of isotopes in soil and xylem water, especially at a large geographical scale, to further understand the long-term changes in plant water use strategy and evaluate their adaptability under climate change. In addition, we observed the rapid appearance of D signals in xylem following tracer

injection, indicating the exchange of bound and mobile water pools in soil and challenging the 'two water world hypothesis'. These findings provide an important reference for evaluating the validity of this hypothesis in semi-arid areas.

**4.4 Limitations due to the extraction method**

In this study, isotopic offset between xylem and soil water was observed for both 11- and 17-year-old unlabeled apple trees (Fig.7 and Table S2). We used the isotopic composition of soil water to correct $\delta$D values of xylem water, ensuring they
match those of soil water. Although we did not collect soil water isotope samples in the isotopic labeling experiments, this may have little effect on determining the soil layer depths from which trees derive their water source due to the high $\delta$D values in the injected solution. It should be noted that isotopic spatial heterogeneity of xylem water induced by sampling position and time (Nehemy et al., 2022) and soil water induced by uneven distribution of throughfall and preferential flow (Xiang et al., 2019; Yang and Fu, 2017) could lead to an isotopic offset. Recently, isotopic offsets between plants and their
potential water sources have been also found in various ecosystems, which may hinder the unambiguous identification of water sources and influence the accurate assessment of DLSW utilization (Barbeta et al., 2022; De La Casa et al., 2022; Zhao et al., 2016). Some studies argued that isotopic fractionation during root water uptake could be attributed to the existence of Casparian strips which can lead to isotope enrichment in root water and depletion in xylem water (Naseer et al., 2012; Vargas et al., 2017). Seeger and Weiler (2021) questioned whether xylem water was completely renewed by newly
absorbed soil water, thus affecting isotopic offset. Furthermore, CVD may mask or exaggerate the isotopic offset, although it was the most common methodology (Chen et al., 2020; Orlowski et al., 2016a,b, 2018; Wen et al., 2022). When quantifying the water use strategies of plants, the isotopic measurement bias related to CVD should be considered. As a whole, there are various trends and causes of isotopic offset; further research about offset is urgently needed to better understand root water uptake processes.

**5 Conclusions**

This study investigated the water use strategy of apple trees using a method combining isotopic labeling and the natural abundance of stable water isotopes and compared the results with the natural abundance of stable water isotopes method. We found that 11- and 17-year-old apple trees had similar water use strategies, switching their main water source from deep to shallow soils based on variation in water availability as the growing season progressed. Overall, apple trees absorbed the
most water from deep (>140 cm) soils during the BYF stage, and 17-year-old apple trees consumed more water in these layers than 11-year-old trees. In addition, the results using the natural abundance of stable water isotopes method clearly overestimated the contribution of DLSW, especially in 320–500 cm soils. Our results highlight that determining whether root water uptake occurs in deep soils will help quantify plant water use strategy and provide insights into the hydrological cycles in DVZ regions. These findings could help understand soil water use strategy of apple trees and hence are favorable to
improve water management practice of apple orchards on the Loess Plateau and maybe other similar areas.

## Data availability

The data that support the findings of this study has been made publicly available in Zenodo (https://doi.org/10.5281/zenodo.7169689).

## Author contributions

XG and XZ conceived the study; SW, GH, XG, MY and XS performed field experiments and collected the data; SW performed the analysis and prepared the first draft of the manuscript; XG, PW and KS edited and commented on the manuscript.

## Competing interests

The authors declare that they have no conflict of interest.

## Acknowledgements


The authors thank Jingjing Jin and Hui Li, Institute of Water-saving Agriculture in Arid Areas of China, Northwest A&F University, for their technical help. This work was jointly supported by the National Key Research and Development Program of China (2021YFD1900700), National Natural Science Foundation of China (42125705), Shaanxi Key Research and Development Program (2020ZDLNY07-04, 2022NY-064), Natural Science Basic Research Program of Shaanxi 410 (2021JC-19), Cyrus Tang Foundation, Chinese Universities Scientific Fund (2452020242) and the 111 Project (grant no. B12007).

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
