# Peer review of "The natural abundance of stable water isotopes method may overestimate deep-layer soil water use by trees"

_Hydrology and Earth System Sciences, 2022_

## Author Comment (AC1)

**Supplement to the response to the reviewer comments**

**Table 1**. General information on the two apple orchards.

| Stand age (a) | Longitude | Latitude | Altitude (m) | Height (cm) | Trunk diameter*(cm) | Crown size (cm) |
|---|---|---|---|---|---|---|
| 11 | 109 °50'13" | 35 °20'5" | 863.8 | 355 | 12.0 | 405×352 |
| 17 | 109 °50'18" | 35 °19'58" | 862.9 | 395 | 14.4 | 450×380 |

[Figure]

**Figure 2:** Time series of meteorological data in 2019 and monthly precipitation in 2019 and multi-year mean, respectively.

[Figure]

**Figure 5:** Temporal dynamics of $\delta$D values in xylem water for 11- and 17-year-old apple trees. Sample collection started on day 1 before $D_2O$ injection and commenced until day 7. Gray dashed lines represent 2SD above the background value (mean $\delta$D values in xylem water on day 1 before labeling).

---

## Author Comment (AC2)

**General comment**

The authors of this manuscript designed a tracer experiment based on stable isotopes of hydrogen and oxygen to identify the depth of soil water taken up by apple trees at three different stages (blossom and young fruit, fruit swelling and fruit maturation stage). The topic of this manuscript is timely and potentially interesting for the readers of Hydrology and Earth System Sciences. Overall, the manuscript is well written and structured, but I found that various and important methodological details (e.g., a detailed description of the extraction method used for soil and vegetation material, and the isotopic composition of the injected water) were not described in the manuscript. Furthermore, the authors should also have considered in the introduction and the discussion recent literature on isotopic fractionation and offset which presents the current technical limitations for the application of stable isotopes in ecohydrological studies. Finally, a section about the limitation of the methodological approach should be added before the conclusions.

**Response:** Thank you for your constructive and encouraging comments and giving us an opportunity to revise this paper. Corrections have been made based on the recommendations, with detailed response to each comment presented below.

(1) Methodological details will be added to the revised manuscript.

"A long polyvinyl chloride pipe was inserted into the holes at the target depth before injecting 300 mL tracer solution ($\delta$D = 714,000‰, 30 mL 99.99% $D_2O$ plus 270 mL tap water) into each hole. The total amount of injected solution was 1,200 mL for each tree.."

"A cryogenic vacuum distillation system (Li-2000; LICA United Technology Limited, Beijing, China) was used to extract water under a heating temperature of 95 ℃ and a pressure of 0.2 Pa, which has been applied in various studies (Huo et al., 2020; Tao et al., 2021a; Wang et al., 2021b; Zhao e al., 2020). The extraction time of soil water and xylem water samples were 90 min and 120 min respectively. Samples were weighed before and after extraction and again after oven-drying for 24 h to calculate the extraction efficiency (Wang et al., 2021), which should be not less than 98%."

**References**

Huo, G., Zhao, X., Gao, X., and Wang, S.: Seasonal effects of intercropping on tree water use strategies in semiarid plantations: Evidence from natural and labelling stable isotopes, Plant Soil, 10.1007/s11104-020-04477-5, 2020.

Tao, Z., Neil, E., and Si, B.: Determining deep root water uptake patterns with tree age in the Chinese loess area, Agric. Water Manage., 249, 106810, https://doi.org/10.1016/j.agwat.2021.106810, 2021.

Wang, H., Jin, J., Cui, B., Si, B., Ma, X., and Wen, M.: Technical note: Evaporating water is

different from bulk soil water in delta $\delta^2H$ and $\delta^{18}O$ and has implications for evaporation calculation, Hydrol. Earth Syst. Sci., 25, 5399-5413, 10.5194/hess-25-5399-2021, 2021.

Zhao, Y., Wang, Y., He, M., Tong, Y., Zhou, J., Guo, X., Liu, J., and Zhang, X.: Transference of *Robinia pseudoacacia* water-use patterns from deep to shallow soil layers during the transition period between the dry and rainy seasons in a water-limited region, For. Ecol. Manage., 457, 10.1016/j.foreco.2019.117727, 2020.

(2) Information on isotopic fractionation will be added to the Introduction and Discussion.

**Introduction**

"Although some recent studies found isotopic fractionation along the soil–root–stem–twig–leaf pathway(e.g., Barbeta et al., 2019; Poca et al., 2019; Vargas et al., 2017), the factors affecting such fractionation remain unclear (Barbeta et al., 2022; Zhao et al., 2016). Orlowski et al. (2016a,b, 2018) suggested that the fractionation is mainly related to cryogenic vacuum distillation (CVD), yet CVD is the most common methodology for water extraction to date (De La Casa et al., 2022)."

**Discussion**

"Isotopic offsets between plants and their potential water sources have been found in various ecosystems, which may hinder the unambiguous identification of water sources and influence the accurate assessment of DLSW utilization (Barbeta et al., 2022; De La Casa et al., 2022; Zhao et al., 2016). Some studies found that isotopic fractionation during root water uptake could be attributed to the existence of Casparian strips, resulting in isotope enrichment in root water and depletion in xylem water (Naseer et al., 2012; Vargas et al., 2017). Seeger and Weiler (2021) suggested whether xylem water was completely renewed by newly absorbed soil water is another important factor affecting isotopic offset."

**References**

Barbeta A., Burlett R., Martín-Gómez P., Fréjaville B., Devert N., Wingate L., Domec J.-C., Ogée J.: Evidence for distinct isotopic compositions of sap and tissue water in tree stems: consequences for plant water source identification, New Phytol., 233, 1121-1132, DOI:10.1111/nph.17857, 2022.

Barbeta A., Jones S.P., Clavé L.,Wingate L., Gimeno T.E., Fréjaville B., Wohl S., Ogée J.: Unexplained hydrogen isotope offsets complicate the identification and quantification of tree water sources in a riparian forest, Hydrol. Earth Syst. Sci., 23, 2129-2146, DOI:10.5194/hess-23-2129-2019, 2019.

De la Casa, J., Barbeta, A., Rodriguez-Una, A., Wingate, L., Ogee, J., and Gimeno, T. E.: Isotopic offsets between bulk plant water and its sources are larger in cool and wet environments, Hydrol. Earth Syst. Sci., 26, 4125-4146, 10.5194/hess-26-4125-2022, 2022.

Naseer, S., Lee, Y., Lapierre, C., Franke, R., Nawrath, C., and Geldner, N.: Casparian strip diffusion barrier in Arabidopsis is made of a lignin polymer without suberin, Proc. Natl. Acad.

Sci. U. S. A., 109(25), 10101-10106, https://doi.org/10.1073/pnas.1205726109, 2012.

Orlowski N., Breuer L., McDonnell J.J.: Ecohydrology Bearings – Invited Commentary Critical issues with cryogenic extraction of soil water for stable isotope analysis, Ecohydrology, 9, 3-10, DOI:10.1002/eco.1722, 2016a.

Orlowski N., Pratt D.L., McDonnell J.J.: Intercomparison of soil pore water extraction methods for stable isotope analysis, Hydrol. Process., 30, 3434-3449, DOI:10.1002/hyp.10870, 2016b.

Orlowski N., Breuer L., Angeli N., Boeckx P., Brumbt C., Cook C.S., Dubbert M., Dyckmans J., Gallagher B., Gralher B., Herbstritt B., Hervé-Fernández P., Hissler C., Koeniger P., Legout A., Macdonald C.J., Oyarzún C., Redelstein R., Seidler C., Siegwolf R., Stumpp C., Thomsen S., Weiler M., Werner C., McDonnell J.J.: Inter-laboratory comparison of cryogenic water extraction systems for stable isotope analysis of soil water, Hydrol. Earth Syst. Sci., 22, 3619-3637, DOI:10.5194/hess-22-3619-2018, 2018.

Poca M., Coomans O., Urcelay C., Zeballos S.R., Bodé S., Boeckx P.: Isotope fractionation during root water uptake by Acacia caven is enhanced by arbuscular mycorrhizas, Plant Soil, 441, 485-497, DOI:10.1007/s11104-019-04139-1, 2019.

Seeger, S. and Weiler, M.: Temporal dynamics of tree xylem water isotopes: in situ monitoring and modeling, Biogeosciences, 18, 4603-4627, 10.5194/bg-18-4603-2021, 2021.

Vargas A.I., Schaffer B., Yuhong L., da Silveira Lobo Sternberg L.: Testing plant use of mobile vs immobile soil water sources using stable isotope experiments, New Phytol., 215, 582-594, DOI:10.1111/nph.14616, 2017.

Zhao L., Wang L., Cernusak L.A., Liu X., Xiao H., Zhou M., Zhang S.: Significant difference in hydrogen isotope composition between xylem and tissue water in *Populus Euphratica*, Plant Cell Environ., 39, 1848-1857, DOI:10.1111/pce.12753, 2016.

(3) The following section will be added to the revised manuscript.

**"4.4 Uncertainty caused by isotopic offset"**

"Isotopic offsets between plants and their potential water sources have been found in various ecosystems, which may hinder the unambiguous identification of water sources and influence the accurate assessment of DLSW utilization (Barbeta et al., 2022; De La Casa et al., 2022; Zhao et al., 2016). Some studies found that isotopic fractionation during root water uptake could be attributed to the existence of Casparian strips, resulting in isotope enrichment in root water and depletion in xylem water (Naseer et al., 2012; Vargas et al., 2017). Seeger and Weiler (2021) suggested whether xylem water was completely renewed by newly absorbed soil water is another important factor affecting isotopic offset. In this study, isotopic offset between xylem and soil water was observed for both 11- and 17-year-old unlabeled apple trees (Fig.7 and Table S2). We used the isotopic composition of soil water to correct $\delta$D values of xylem water, ensuring they match those of soil water. In comparison, although we did not collect soil water isotope samples in the isotope labeling experiments, and thus could not correct the $\delta$D values in xylem water of labeled apple trees, this may have little effect on

determining the soil layer depths from which trees derive their water source due to the high $\delta$D values in the injected solution. It should be noted that isotopic spatial heterogeneity related to destructive sampling (xylem and soil water) could lead to an isotopic mismatch between xylem and soil water. In addition, CVD may mask or exaggerate the isotopic offset, although it was the most common methodology (Orlowski et al., 2016a,b, 2018). When quantifying the water use strategies of plants, the isotopic measurement bias related to CVD should be considered. As a whole, there are various trends and causes of isotopic offset; further research about offset is urgently needed to better understand root water uptake processes."

**References**

Barbeta A., Burlett R., Martín-Gómez P., Fréjaville B., Devert N., Wingate L., Domec J.-C., Ogée J.: Evidence for distinct isotopic compositions of sap and tissue water in tree stems: consequences for plant water source identification, New Phytol., 233, 1121-1132, DOI:10.1111/nph.17857, 2022.

De la Casa, J., Barbeta, A., Rodriguez-Una, A., Wingate, L., Ogee, J., and Gimeno, T. E.: Isotopic offsets between bulk plant water and its sources are larger in cool and wet environments, Hydrol. Earth Syst. Sci., 26, 4125-4146, 10.5194/hess-26-4125-2022, 2022.

Naseer, S., Lee, Y., Lapierre, C., Franke, R., Nawrath, C., and Geldner, N.: Casparian strip diffusion barrier in Arabidopsis is made of a lignin polymer without suberin, Proc. Natl. Acad. Sci. U. S. A., 109(25), 10101-10106, https://doi.org/10.1073/pnas.1205726109, 2012.

Orlowski N., Breuer L., McDonnell J.J.: Ecohydrology Bearings – Invited Commentary Critical issues with cryogenic extraction of soil water for stable isotope analysis, Ecohydrology, 9, 3-10, DOI:10.1002/eco.1722, 2016a.

Orlowski N., Pratt D.L., McDonnell J.J.: Intercomparison of soil pore water extraction methods for stable isotope analysis, Hydrol. Process., 30, 3434-3449, DOI:10.1002/hyp.10870, 2016b.

Orlowski N., Breuer L., Angeli N., Boeckx P., Brumbt C., Cook C.S., Dubbert M., Dyckmans J., Gallagher B., Gralher B., Herbstritt B., Hervé-Fernández P., Hissler C., Koeniger P., Legout A., Macdonald C.J., Oyarzún C., Redelstein R., Seidler C., Siegwolf R., Stumpp C., Thomsen S., Weiler M., Werner C., McDonnell J.J.: Inter-laboratory comparison of cryogenic water extraction systems for stable isotope analysis of soil water, Hydrol. Earth Syst. Sci., 22, 3619-3637, DOI:10.5194/hess-22-3619-2018, 2018.

Seeger, S. and Weiler, M.: Temporal dynamics of tree xylem water isotopes: in situ monitoring and modeling, Biogeosciences, 18, 4603-4627, 10.5194/bg-18-4603-2021, 2021.

Vargas A.I., Schaffer B., Yuhong L., da Silveira Lobo Sternberg L.: Testing plant use of mobile vs immobile soil water sources using stable isotope experiments, New Phytol., 215, 582-594, DOI:10.1111/nph.14616, 2017.

Zhao L., Wang L., Cernusak L.A., Liu X., Xiao H., Zhou M., Zhang S.: Significant difference in hydrogen isotope composition between xylem and tissue water in *Populus Euphratica*, Plant Cell Environ., 39, 1848-1857, DOI:10.1111/pce.12753, 2016.

**Specific comments**

**Lines 44-45:** The authors should mention in the introduction that isotopic fractionation has been observed in various studies (e.g., Poca et al., 2019; Vargas et al., 2017; Barbeta et al., 2019), along the soil-root-stem-twig-leaf pathway. The factors affecting such fractionation/offset are still unclear, but they seem to be mainly related to the water extraction technique, and particularly to cryogenic vacuum distillation (e.g., Zhao et al., 2016; Barbeta et al., 2022).

**Response:** Agreed. Information on isotopic fractionation will be added to the Introduction.

"Although some recent studies found isotopic fractionation along the soil–root–stem–twig–leaf pathway(e.g., Barbeta et al., 2019; Poca et al., 2019; Vargas et al., 2017), the factors affecting such fractionation remain unclear (Barbeta et al., 2022; Zhao et al., 2016). Orlowski et al. (2016a,b, 2018) suggested that the fractionation is mainly related to cryogenic vacuum distillation (CVD), yet CVD is the most common methodology for water extraction to date (De La Casa et al., 2022)."

**References**

Barbeta A., Burlett R., Martín-Gómez P., Fréjaville B., Devert N., Wingate L., Domec J.-C., Ogée J.: Evidence for distinct isotopic compositions of sap and tissue water in tree stems: consequences for plant water source identification, New Phytol., 233, 1121-1132, DOI:10.1111/nph.17857, 2022.

Barbeta A., Jones S.P., Clavé L.,Wingate L., Gimeno T.E., Fréjaville B., Wohl S., Ogée J.: Unexplained hydrogen isotope offsets complicate the identification and quantification of tree water sources in a riparian forest, Hydrol. Earth Syst. Sci., 23, 2129-2146, DOI:10.5194/hess-23-2129-2019, 2019.

De la Casa, J., Barbeta, A., Rodriguez-Una, A., Wingate, L., Ogee, J., and Gimeno, T. E.: Isotopic offsets between bulk plant water and its sources are larger in cool and wet environments, Hydrol. Earth Syst. Sci., 26, 4125-4146, 10.5194/hess-26-4125-2022, 2022.

Orlowski N., Breuer L., McDonnell J.J.: Ecohydrology Bearings – Invited Commentary Critical issues with cryogenic extraction of soil water for stable isotope analysis, Ecohydrology, 9, 3-10, DOI:10.1002/eco.1722, 2016a.

Orlowski N., Pratt D.L., McDonnell J.J.: Intercomparison of soil pore water extraction methods for stable isotope analysis, Hydrol. Process., 30, 3434-3449, DOI:10.1002/hyp.10870, 2016b.

Orlowski N., Breuer L., Angeli N., Boeckx P., Brumbt C., Cook C.S., Dubbert M., Dyckmans J., Gallagher B., Gralher B., Herbstritt B., Hervé-Fernández P., Hissler C., Koeniger P., Legout A., Macdonald C.J., Oyarzún C., Redelstein R., Seidler C., Siegwolf R., Stumpp C., Thomsen S., Weiler M., Werner C., McDonnell J.J.: Inter-laboratory comparison of cryogenic water extraction systems for stable isotope analysis of soil water, Hydrol. Earth Syst. Sci., 22, 3619-3637, DOI:10.5194/hess-22-3619-2018, 2018.

Poca M., Coomans O., Urcelay C., Zeballos S.R., Bodé S., Boeckx P.: Isotope fractionation during root water uptake by Acacia caven is enhanced by arbuscular mycorrhizas, Plant Soil,

441, 485-497, DOI:10.1007/s11104-019-04139-1, 2019.

Vargas A.I., Schaffer B., Yuhong L., da Silveira Lobo Sternberg L.: Testing plant use of mobile vs immobile soil water sources using stable isotope experiments, New Phytol., 215, 582-594, DOI:10.1111/nph.14616, 2017.

Zhao L., Wang L., Cernusak L.A., Liu X., Xiao H., Zhou M., Zhang S.: Significant difference in hydrogen isotope composition between xylem and tissue water in *Populus Euphratica*, Plant Cell Environ., 39, 1848-1857, DOI:10.1111/pce.12753, 2016.

**Lines 50-52:** The authors should mention limitations due to the extraction technique applied to soils (e.g., cryogenic vacuum distillation, Orlowski et al. (2016a,b, 2018)).

**Response:** Agreed. We will change the manuscript as follows.

"Orlowski et al. (2016a,b, 2018) suggested that the fractionation is mainly related to cryogenic vacuum distillation (CVD), yet CVD is the most common methodology for water extraction to date (De La Casa et al., 2022)."

**References**

De la Casa, J., Barbeta, A., Rodriguez-Una, A., Wingate, L., Ogee, J., and Gimeno, T. E.: Isotopic offsets between bulk plant water and its sources are larger in cool and wet environments, Hydrol. Earth Syst. Sci., 26, 4125-4146, 10.5194/hess-26-4125-2022, 2022.

Orlowski N., Breuer L., McDonnell J.J.: Ecohydrology Bearings – Invited Commentary Critical issues with cryogenic extraction of soil water for stable isotope analysis, Ecohydrology, 9, 3-10, DOI:10.1002/eco.1722, 2016a.

Orlowski N., Pratt D.L., McDonnell J.J.: Intercomparison of soil pore water extraction methods for stable isotope analysis, Hydrol. Process., 30, 3434-3449, DOI:10.1002/hyp.10870, 2016b.

Orlowski N., Breuer L., Angeli N., Boeckx P., Brumbt C., Cook C.S., Dubbert M., Dyckmans J., Gallagher B., Gralher B., Herbstritt B., Hervé-Fernández P., Hissler C., Koeniger P., Legout A., Macdonald C.J., Oyarzún C., Redelstein R., Seidler C., Siegwolf R., Stumpp C., Thomsen S., Weiler M., Werner C., McDonnell J.J.: Inter-laboratory comparison of cryogenic water extraction systems for stable isotope analysis of soil water, Hydrol. Earth Syst. Sci., 22, 3619-3637, DOI:10.5194/hess-22-3619-2018, 2018.

**Lines 90-92:** What is the depth to the water table in the study area? Do the roots of the apple trees have access to shallow groundwater?

**Response:** The groundwater level in the study area is over 100 m deep on average, which cannot be reached by plant roots. We will amend the manuscript.

**Lines 92-93:** What is the average, minimum and maximum distance between the trees?

Does the tree distance affect the root density and possibly the water uptake?

**Response:**

(1) The plant and row spacing for the two orchards was 4 × 4 m, with all trees planted at the same interval.

(2) The results of two-dimensional root distribution of apple tree roots from Huo et al. (2021) showed that apple trees in shallow soil layers had relatively high root density and wide horizontal root distribution. With soil depth and horizontal distance increasing, the root density of apple trees significantly decreased. The tree root density was low and horizontal distribution was narrow in deep soil layers. Therefore, tree distance had little effect on root density and water uptake.

We will make changes in the revised manuscript.

**References**

Huo, G., Gosme, M., Gao, X., Dupraz, C., Yang, J., and Zhao, X.: Dynamics of interspecific water relationship in vertical and horizontal dimensions under a dryland apple-*Brassica* intercropping Quantifying by experiments and the 3D Hi-sAFe model, Agr. Forest Meteorol., 310, 108620, 10.1016/j.agrformet.2021.108620, 2021.

**Table 1:** It is unclear whether height, trunk and crown size data represent an average or single values. The authors should add the minimum and maximum values for all characteristics, as well as the number of trees used in this study.

**Response:** The height, trunk diameter (TD) and crown size (CS) were mean values for 20 trees in each orchard. The maximum and minimum values for all characteristics will be shown in Table S1.

**Table S1** The maximum and minimum values of the height (H), trunk diameter (TD) and crown size (CS) for apple trees in two orchard.

| Stand age (a) | $H_{max}$ (cm) | $H_{min}$ (cm) | $TD_{max}$ (cm) | $TD_{min}$ (cm) | $CS_{max}$ (cm) | $CS_{min}$ (cm) |
|---|---|---|---|---|---|---|
| 11 | 403 | 320 | 12.8 | 11.5 | 430 × 400 | 360 × 320 |
| 17 | 450 | 355 | 15.2 | 13.8 | 475 × 420 | 390 × 340 |

**Lines 108-109:** The authors should report the isotopic composition of the water mixture used for injection/irrigation, as well as the total amount of irrigation water applied per each tree.

**Response:** The isotopic composition of the tracer solution and total amount of injected solution will be added to the revised manuscript.

"A long polyvinyl chloride pipe was inserted into the holes at the target depth before injecting 300 mL tracer solution ($\delta$D = 714,000‰, 30 mL 99.99% $D_2O$ plus 270 mL

tap water) into each hole. The total amount of injected solution was 1,200 mL for each tree."

**Lines 109-111:** I do not understand the aim of this sentence. Based on it, I understand that the authors may have injected a water amount that could be too small to detect a soil water content variation and perhaps, also an isotopic variation.

**Response:**

(1) The results from Huo et al. (2020) showed that 300 mL of aqueous solution wet 400 $cm^3$ of the soil on the Loess Plateau, equivalent to a < 1% change in SWC (within the measurement error), so the impact on soil hydrological processes was negligible.

(2) Unlike soil water content, the isotopic composition of soil water changed obviously after injection due to the high isotopic value of D (714,000‰) in the injected solution, which could reach to 150000‰.

**Lines 112-113:** Details about the sample size should be reported in the text here or in an additional table. Furthermore, more details are needed to understand how the background value was computed (is it an average of how many samples?), and where these unlabeled trees are located compared to the trees used for the experiment.

**Response:** Agreed. Two xylem samples were collected for each tree, with a total sample size of six for each treatment in a single sampling. The background value was an average of six xylem samples from unlabeled trees. The layout of unlabeled trees and labeled trees will be shown in Figure S1.

[Figure]

**Figure S1** The layout of apple trees in experiment plot.

**Lines 132-133:** I am not familiar with this cryogenic vacuum distillation system, and I am not sure it is manufactured by Los Gatos Research. I recommend adding here a detailed description of the extraction method, as well as information about the extraction efficiency.

**Response:**

(1) The manufacturer of the cryogenic vacuum distillation system will be revised.

"A cryogenic vacuum distillation system (Li-2000; LICA United Technology Limited, Beijing, China) was used to extract water under a heating temperature of 95 ℃ and a pressure of 0.2 Pa, which has been applied in various studies (Huo et al., 2020; Tao et al., 2021a; Wang et al., 2021b; Zhao e al., 2020)."

(2) A detailed description of the cryogenic vacuum distillation method and its extraction efficiency will be added to the revised manuscript.

"A cryogenic vacuum distillation system (Li-2000; LICA United Technology Limited, Beijing, China) was used to extract water under a heating temperature of 95 ℃ and a pressure of 0.2 Pa, which has been applied in various studies (Huo et al., 2020; Tao et al., 2021a; Wang et al., 2021b; Zhao e al., 2020). The extraction time of soil water and xylem water samples were 90 min and 120 min respectively. Samples were weighed before and after extraction and again after oven-drying for 24 h to calculate the extraction efficiency (Wang et al., 2021b), which should be not less than 98%."

**Reference**

Huo, G., Zhao, X., Gao, X., and Wang, S.: Seasonal effects of intercropping on tree water use strategies in semiarid plantations: Evidence from natural and labelling stable isotopes, Plant Soil, 10.1007/s11104-020-04477-5, 2020.

Tao, Z., Neil, E., and Si, B.: Determining deep root water uptake patterns with tree age in the Chinese loess area, Agric. Water Manage., 249, 106810, https://doi.org/10.1016/j.agwat.2021.106810, 2021b.

Wang, H., Jin, J., Cui, B., Si, B., Ma, X., and Wen, M.: Technical note: Evaporating water is different from bulk soil water in delta $\delta^2$H and $\delta^{18}$O and has implications for evaporation calculation, Hydrol. Earth Syst. Sci., 25, 5399-5413, 10.5194/hess-25-5399-2021, 2021.

Zhao, Y., Wang, Y., He, M., Tong, Y., Zhou, J., Guo, X., Liu, J., and Zhang, X.: Transference of *Robinia pseudoacacia* water-use patterns from deep to shallow soil layers during the transition period between the dry and rainy seasons in a water-limited region, For. Ecol. Manage., 457, 10.1016/j.foreco.2019.117727, 2020.

**Lines 133-135:** Details about the uncertainty in the isotopic analyses for each instrument should be added here. Did the authors use specific practices to mitigate the memory effect in the isotopic measurements?

**Response:**

(1) The measurement precision of $\delta^{18}$O and $\delta$D is 0.2‰ and 1.0‰ for the TIWA-45EP isotope ratio infrared spectroscopy analyzer and 0.3‰ and 2.0‰ for the Stable Isotope Ratio Mass Spectrometer, respectively.

(2) Yes. "Each isotopic sample was repeatedly measured six times. The first three measurements were discarded to mitigate the memory effect of isotopic measurement, and the mean value of the last three measurements was taken as the isotopic value of sample."

The relevant content will be added to the revised manuscript.

**Line 143:** Considering recent literature, the no isotopic fractionation assumption is a strong assumption that should be tested. Did the authors check whether their isotopic data present an offset compared to the isotopic composition of local water sources (e.g.,

precipitation, soil water before the tracer experiment and shallow groundwater) and the water used for the tracer experiment?

**Response:** In general, soil water is the primary water source for trees on the Loess Plateau. We assessed the isotopic offset between xylem water and soil water using soil water line conditioned excess (SW-excess) proposed by Barbeta et al. (2019):

$$\text{SW-excess} = \delta D - a_s \delta^{18}O - b_s \tag{1}$$

where $a_s$ and $b_s$ are the slope and intercept of soil water line (SWL), respectively; $\delta D$ and $\delta^{18}O$ are the isotopic compositions of xylem water. A positive SW-excess value means that xylem water plots above SWL in a $\delta D - \delta^{18}O$ diagram (i.e. D in xylem water is more enriched than SWL), while a negative value means that xylem water plots below SWL in a $\delta D - \delta^{18}O$ diagram (i.e. D in xylem water is more depleted than SWL).

In the Bayesian isotope mixing model, the $\delta D$ and $\delta^{18}O$ values for each potential water source were used as source data; the $\delta D$ after subtracting the SW-excess and $\delta^{18}O$ values for xylem water were used as mixture data. $\delta D$ values of xylem water corrected by SWL can match those of soil water. Thus, the fractionation factor in this model was set to zero, assuming no isotope fractionation during root water uptake.

We will make changes to the revised manuscript.

**References**

Barbeta A., Jones S.P., Clavé L.,Wingate L., Gimeno T.E., Fréjaville B., Wohl S., Ogée J.: Unexplained hydrogen isotope offsets complicate the identification and quantification of tree water sources in a riparian forest, Hydrol. Earth Syst. Sci., 23, 2129-2146, DOI:10.5194/hess-23-2129-2019, 2019.

**Lines 145-148:** I do not understand the purpose of this index, S. The description of this index should be improved, e.g. by adding what positive and negative values indicate, and the ranges used for the soil water content.

**Response:** The index S represented the sensibility of the response of root water uptake to soil water content increase. A higher S value means a faster response of root water uptake and greater sensitivity to moisture changes. As the index is likely to lead to misunderstanding, it will be deleted in the revised manuscript.

**Figure 5:** The isotopic composition of the injected water should be plotted, and I suggest showing the background value also without the 2 SD. Furthermore, the sample size should be reported in the caption.

**Response:**

(1) Because the concentration of D in tracer solution ($\delta D = 714,000‰$) is much higher than it in xylem water, it cannot be shown in the Figure 5. We will add the isotopic

composition of the tracer solution to Figure 5's title.

(2) The background value without 2 SD will be added to Figure 5.

(3) The sample size will be added to Figure 5's title.

[Figure]

**Figure 5:** Temporal dynamics of $\delta$D values in xylem water for 11- and 17-year-old apple trees. Sample collection started on day 1 before D$_2$O tracer solution ($\delta$D = 714000‰) injection and commenced until day 7 (N=6). Gray dashed lines represent the background value (mean $\delta$D values in xylem water on day 1 before labeling) and black dashed lines represent 2 SD above the background value.

**Figure 7:** In the plots for "FSW for 11yr" and "BYF for 17yr" there may be an isotopic offset for some xylem water samples; I suggest checking whether there is a significant deviation from the soil water isotopic line, by also considering the uncertainty due to the isotopic analyses. In these plots, the authors should add the isotopic composition of the background values (in soil and xylem waters) and of the injected water. Equation of the LMWL and details about sample size and when the samples were collected should be added in the caption.

**Response:**

(1) We assessed the isotopic offset between xylem water and soil water using soil water line conditioned excess (SW-excess) proposed by Barbeta et al. (2019):

$$SW\text{-excess} = \delta D - a_s\delta^{18}O - b_s \tag{1}$$

where $a_s$ and $b_s$ are the slope and intercept of soil water line (SWL), respectively; $\delta D$ and $\delta^{18}O$ are the isotopic compositions of xylem water.

**Table S2** Mean (±SD, n=6) soil water excess (SW-excess, ‰) for 11- and 17-year-old apple trees.

| Stand age (a) | Growth stage | | |
|---|---|---|---|
| | BYF | FSW | FTM |
| 11 | 2.09±0.83 | -6.80±3.56 | 4.78±1.85 |
| 17 | -2.75±075 | -2.57±0.29 | -0.80±1.72 |

The results showed that SW-excess was negative in FSW stage for 11yr and BYF stage for 17yr trees, which meant that xylem water plots below SWL in a $\delta D - \delta^{18}O$ diagram (i.e. D in xylem water is more depleted than SWL). Therefore, the $\delta D$ after subtracting the SW-excess and $\delta^{18}O$ values for xylem water were used as mixture data for the Bayesian isotope mixing model.

(2) The soil water line and xylem water line will be added to Figure 7. We did not show the isotopic composition of injected water because the soil water and xylem water in these plots were from unlabeled trees.

[Figure]

**Figure 7:** $\delta D$ and $\delta^{18}O$ values in xylem water and different soil layers (0–40 cm, 40–140 cm, 140–240 cm, 240–320 cm) for 11- and 17-year-old apple trees (±SD). The GMWL and LMWL represents the global and local meteoric water lines. LMWL: $\delta D = 7.1\delta^{18}O+2.1$. The SWL and

XWL represents the soil water line and xylem water line.

(3) The LMWL Equation will be added. "LMWL: $\delta D = 7.1\delta^{18}O + 2.1$".

Details on sample size and sample timing will be added to the "Materials and Methods" and Figure 2.

"Rain samples (N = 32) were collected using a combined device of polyethylene bottle and funnel during rainfall events between May and September. A plastic ball was placed on the funnel to prevent evaporation. The collected rainwater samples were immediately sealed into vials by parafilm and stored at 4°C for isotopic determination."

[Figure]

**Figure 2:** Time series of meteorological data and rainwater isotopic values in 2019 and monthly precipitation in 2019 and multi-year mean, respectively.

**Figure 9:** Please remove the regression line when there are only three samples used for the analysis.

**Response:** Agreed. The regression line will be removed in the revised manuscript.

[Figure]

**Figure 9:** Relationship between the contribution of water sources and soil water content in different soil layers of 11- and 17-year-old apple orchards.

**Section 3.6:** I expected to read here the results concerning the application of the index S, but they are not reported. Therefore, I suggest adding here such results, or removing the description of S at Lines 145-148.

**Response:** Agreed. The description of S will be deleted in the revised manuscript.

**Section 4.3:** Please add in this section (or in a new one) the limitations of the experimental approach. I think the limitations of this work are mainly related to the assumptions of no isotopic fractionation and negligible spatial variability in the isotopic composition of unlabeled and labeled trees, and to the extraction method (cryogenic

vacuum distillation system).

**Response:** We will add a new section to the revised manuscript.

**"4.4 Uncertainty caused by isotopic offset"**

"Isotopic offsets between plants and their potential water sources have been found in various ecosystems, which may hinder the unambiguous identification of water sources and influence the accurate assessment of DLSW utilization (Barbeta et al., 2022; De La Casa et al., 2022; Zhao et al., 2016). Some studies found that isotopic fractionation during root water uptake could be attributed to the existence of Casparian strips, resulting in isotope enrichment in root water and depletion in xylem water (Naseer et al., 2012; Vargas et al., 2017). Seeger and Weiler (2021) suggested whether xylem water was completely renewed by newly absorbed soil water is another important factor affecting isotopic offset. In this study, isotopic offset between xylem and soil water was observed for both 11- and 17-year-old unlabeled apple trees (Fig.7 and Table S2). We used the isotopic composition of soil water to correct $\delta D$ values of xylem water, ensuring they match those of soil water. In comparison, although we did not collect soil water isotope samples in the isotope labeling experiments, and thus could not correct the $\delta D$ values in xylem water of labeled apple trees, this may have little effect on determining the soil layer depths from which trees derive their water source due to the high $\delta D$ values in the injected solution. It should be noted that isotopic spatial heterogeneity related to destructive sampling (xylem and soil water) could lead to an isotopic mismatch between xylem and soil water. In addition, CVD may mask or exaggerate the isotopic offset, although it was the most common methodology (Orlowski et al., 2016a,b, 2018). When quantifying the water use strategies of plants, the isotopic measurement bias related to CVD should be considered. As a whole, there are various trends and causes of isotopic offset; further research about offset is urgently needed to better understand root water uptake processes."

**References**

Barbeta A., Burlett R., Martín-Gómez P., Fréjaville B., Devert N., Wingate L., Domec J.-C., Ogée J.: Evidence for distinct isotopic compositions of sap and tissue water in tree stems: consequences for plant water source identification, New Phytol., 233, 1121-1132, DOI:10.1111/nph.17857, 2022.

De la Casa, J., Barbeta, A., Rodriguez-Una, A., Wingate, L., Ogee, J., and Gimeno, T. E.: Isotopic offsets between bulk plant water and its sources are larger in cool and wet environments, Hydrol. Earth Syst. Sci., 26, 4125-4146, 10.5194/hess-26-4125-2022, 2022.

Naseer, S., Lee, Y., Lapierre, C., Franke, R., Nawrath, C., and Geldner, N.: Casparian strip diffusion barrier in Arabidopsis is made of a lignin polymer without suberin, Proc. Natl. Acad. Sci. U. S. A., 109(25), 10101-10106, https://doi.org/10.1073/pnas.1205726109, 2012.

Orlowski N., Breuer L., McDonnell J.J.: Ecohydrology Bearings – Invited Commentary Critical issues with cryogenic extraction of soil water for stable isotope analysis, Ecohydrology, 9, 3-10, DOI:10.1002/eco.1722, 2016a.

Orlowski N., Pratt D.L., McDonnell J.J.: Intercomparison of soil pore water extraction methods for stable isotope analysis, Hydrol. Process., 30, 3434-3449, DOI:10.1002/hyp.10870, 2016b.

Orlowski N., Breuer L., Angeli N., Boeckx P., Brumbt C., Cook C.S., Dubbert M., Dyckmans J., Gallagher B., Gralher B., Herbstritt B., Hervé-Fernández P., Hissler C., Koeniger P., Legout A., Macdonald C.J., Oyarzún C., Redelstein R., Seidler C., Siegwolf R., Stumpp C., Thomsen S., Weiler M., Werner C., McDonnell J.J.: Inter-laboratory comparison of cryogenic water extraction systems for stable isotope analysis of soil water, Hydrol. Earth Syst. Sci., 22, 3619-3637, DOI:10.5194/hess-22-3619-2018, 2018.

Seeger, S. and Weiler, M.: Temporal dynamics of tree xylem water isotopes: in situ monitoring and modeling, Biogeosciences, 18, 4603-4627, 10.5194/bg-18-4603-2021, 2021.

Vargas A.I., Schaffer B., Yuhong L., da Silveira Lobo Sternberg L.: Testing plant use of mobile vs immobile soil water sources using stable isotope experiments, New Phytol., 215, 582-594, DOI:10.1111/nph.14616, 2017.

Zhao L., Wang L., Cernusak L.A., Liu X., Xiao H., Zhou M., Zhang S.: Significant difference in hydrogen isotope composition between xylem and tissue water in *Populus Euphratica*, Plant Cell Environ., 39, 1848-1857, DOI:10.1111/pce.12753, 2016.

**Technical corrections**

**Line 43:** I think the authors should write "Analytical techniques based on stable isotopes…", and furthermore, they should consider that many sampling techniques are destructive because they require the collection of soil and vegetation material (e.g., leaves, twigs, wood cores etc.).

**Response:** Agreed. We will change the revised manuscript.

"Analytical techniques based on stable isotopes ($\delta$D and $\delta^{18}$O) can be applied to study plant water use based on the assumption that no isotope fractionation occurs during root water uptake (Dawson et al., 2002; Ehleringer and Dawson, 1992; Evaristo et al., 2015; Rothfuss and Javaux, 2017)."

"It should be noted that isotopic spatial heterogeneity related to destructive sampling (xylem and soil water) could lead to an isotopic mismatch between xylem and soil water."

**References**

Dawson, T. E., Mambelli, S., Plamboeck, A. H., Templer, P. H., and Tu, K. P.: Stable Isotopes in Plant Ecology, Annu. Rev. Ecol. Syst., 33, 507-559, 10.1146/annurev.ecolsys.33.020602.095451, 2002.

Ehleringer, J. R. and Dawson, T. E.: WATER-UPTAKE BY PLANTS - PERSPECTIVES FROM STABLE ISOTOPE COMPOSITION, Plant Cell Environ., 15, 1073-1082, 10.1111/j.1365-3040.1992.tb01657.x, 1992.

Evaristo, J., Jasechko, S., and McDonnell, J. J.: Global separation of plant transpiration from groundwater and streamflow, Nature, 525, 91-94, 10.1038/nature14983, 2015.

Rothfuss, Y. and Javaux, M.: Reviews and syntheses: Isotopic approaches to quantify root water

uptake: a review and comparison of methods, Biogeosciences, 14, 2199-2224, 10.5194/bg-14-2199-2017, 2017.

**Line 51:** Please replace "confusion of" with "unclear".

**Response:** Agreed. We will change the revised manuscript.

"However, it is challenging to quantify where in the soil profile the roots extract water due to limitations in monitoring technologies and unclear physical processes such as preferential flow."

**Title of section 2.2:** I suggest changing it with "Sample collection".

**Response:** Agreed. We will change the revised manuscript.

"**2.2** Sample collection"

**Title of section 2.2.2:** I suggest changing it with "Collection of soil and vegetation samples for isotopic analysis".

**Response:** Agreed. We will change the revised manuscript.

"2.2.2 Collection of soil and vegetation samples for isotopic analysis"

**Figure 4:** Please add in the caption when the soil water content was determined (before, during or after the tracer injection).

**Response:** Agreed. We will change the revised manuscript.

"Figure 4: Vertical distribution of soil water content (SWC) before the tracer injection in 11-year-old (A) and 17-year-old (B) apple orchards. Values are means ± SD (N=3)."

**Figure 8:** Please remove from the caption "Seasonal patterns of" because the results refer only to three specific tracer injections.

**Response:** Agreed. We will change the revised manuscript.

"**Figure 8:** The contribution of four potential water sources to xylem water in 11-year-old (A) and 17-year-old (B) apple trees. Error bars indicate standard errors of the means (N=3). Asterisks represent significant differences between growing stages (*, $P < 0.05$; **, $P < 0.01$; ***, $P < 0.001$)."

---

## Author Response (AR1)

HESS-2022-142

Title: The natural abundance of stable water isotopes method may overestimate deep-layer soil water use by trees

Author(s): Shaofei Wang et al.

MS type: Research article

Iteration: Major revision

**Comments from handling editor:**

Dear authors,

the comments of the two Referees are somewhat split, but it seems to me that the issues raised by Reviewer #2 are serious and should be carefully addressed.

However, from your response to the comments received, I can see that you are going to thoroughly revise your manuscript.

I look forward to receiving the revised manuscript for further review.

Best regards

Roberto Greco

**Response:** Thank you for your letter and the referees' comments concerning our manuscript entitled "The natural abundance of stable water isotopes method may overestimate deep-layer soil water use by trees" (hess-2022-142). Those comments are valuable and very helpful. We have read through comments carefully and have made corrections. Based on the instructions provided in your letter, we uploaded the file of the revised manuscript. Revisions in the text are shown using red highlight for additions, and strikethrough font for deletions. The responses to the reviewer's comments are marked in blue and presented following.

**Anonymous Referee #1**

With great pleasure I read your manuscript, where you investigate the water use strategy of apple trees by different ages. You injected labeled water ($D_2O$) and studied at which depth the trees withdraw their waters. This analysis was carried out for three different growing stages. The paper is very well structured, easy to read, informative figures and in good English language. The applied method is correct and I have no comments about the conclusion. Hence my comments limit mostly to technical issues, except from the following two comments:

**Response:** Thank you very much for the constructive and encouraging comments and giving us an opportunity to revise this paper. Corrections have been made based on

the recommendations, and the detailed response to each comment is presented as follow.

**Comment 1**

From a scientific point of view your work is really interesting, but the question remains what we can do with this information (social relevance). Furthermore, the study only looks at one growing cycle and ignores that water use strategies change depending on water availability (or climate). In case plants experience dry spells, their roots develop differently in comparison to plant that do not experience dry spells. So there is also a long-term strategy, where plants (sometimes) can adapt to climate change. Could you discuss on this topic?

**Response:** Thanks for your comments. Here are our responses.

(1) On the Loess Plateau, apple trees are the dominant cash tree plantations. Over recent decades, the cultivated area of apple trees increased continuously, with the Plateau becoming the largest apple tree cultivation zone globally, accounting for more than one-quarter of global coverage and production (Gao et al., 2021). The apple industry has become the backbone of the local rural economy, involving more than 10 million farmers (Gao et al., 2021). The apple trees on the Loess Plateau are heavily dependent on soil water in deep layers due to low annual precipitation (400-600 mm) and abundant soil water resource in deep vadose zone (Wang et al., 2021; Yang et al., 2022). Therefore, it is of great significance to determine the utilization time of deep-layer soil water for orchard water management and apple yield improvement. We have added the social relevance of the findings to the revised manuscript (L 83-89 and 364-365).

**References**

[revised manuscript text omitted]

**Comment 2**
Data availability: I don't think the current data-statement is sufficient. Data that is used in publications should preferable be available online and not "upon request". The latter is only possible in exceptional cases. If this is the case, this should be justified.

**Response:** Agreed. The text has been revised (L 419).

"The data that support the findings of this study has been made publicly available in Zenodo (https://doi.org/10.5281/zenodo.7169689)."

**Technical issues:**

**L60:** BYF is note explained in the main text (only in the abstract). I think it's good practice to define abbreviations the first time you mention them in the main text.

**Response:** Agreed. We have amended the manuscript (L 66).

"Wang et al. (2020) argued that the absorption of deep soil water only occurred during the blossom and young fruit (BYF) stage in apple orchards".

**L88**: unit of annual rainfall in mm/y.

**Response:** Done (L 96).

"Mean annual precipitation in the study region is 507.9 mm/y".

**Table1**: I would recommend to change the way the units are provided. I would skip the '/' and use brackets.

**Response:** Agreed. The text has been revised (L 106).

**Table 1. General information on the two apple orchards.**

| Stand age (a) | Longitude | Latitude | Altitude (m) | Height (cm) | Trunk diameter*(cm) | Crown size (cm) |
|---|---|---|---|---|---|---|
| 11 | 109 °50'13" | 35 °20'5" | 863.8 | 355 | 12.0 | 405×352 |
| 17 | 109 °50'18" | 35 °19'58" | 862.9 | 395 | 14.4 | 450×380 |

**L155**-and further: all variables/parameters should be in italic.

**Response:** Done (L 190-192).

"Figure 2 shows the total precipitation ($P_t$) and growing season (April to September) precipitation ($P_g$). $P_t$ and $P_g$ in 2019 were 522.1 mm and 442.3 mm, respectively, similar to the multiyear (1999–2018) mean (507.9 mm/y for $P_t$ and 407.5 mm/y for $P_g$)."

**L156**: "mean annual $P_t$": this is long-term $P_t$? If so, provide period. Furthermore unit should be mm/y.

**Response:** Yes, "mean annual $P_t$" is long-term $P_t$. The period has been added to the revised manuscript (L 191).

"$P_t$ and $P_g$ in 2019 were 522.1 mm and 442.3 mm, respectively, similar to the multiyear (1999–2018) means (507.9 mm/y for $P_t$ and 407.5 mm/y for $P_g$)."

**L156-157**: but the monthly rainfall can differ a lot (see figure2b). So is 2019 a normal year?

**Response:** The study area is located in China's Loess Plateau, its most significant climatological characteristics are distinctly seasonal precipitation, approximately 55-78% of which falls in June through September (Fu et al., 2017; Jia et al., 2017). In 2019, 74.9% of the precipitation in study area fell in June through September, according with the seasonal distribution characteristics of precipitation in the Plateau. In addition, the total precipitation ($P_t$) and growing season (April to September) precipitation ($P_g$) in 2019 were 522.1 mm and 442.3 mm, respectively, similar to the multiyear (1999–2018) mean (507.9 mm/y for $P_t$ and 407.5 mm/y for $P_g$). In this way, the year of 2019 was considered a normal precipitation year. We have made changes in the revised manuscript (L 190-194).

**References**

Fu, B., Wang, S., Liu, Y., Liu, J., Liang, W., and Miao, C.: Hydrogeomorphic Ecosystem Responses to Natural and Anthropogenic Changes in the Loess Plateau of China, Annu. Rev. Earth Pl. Sc., 45(1), 223-243, DOI:10.1146/annurev-earth-063016-020552, 2017.

Jia, X., Shao, M., Zhu, Y., and Luo, Y.: Soil moisture decline due to afforestation across the Loess Plateau, China, J. Hydro., 546, 113-122, DOI:10.1016/j.jhydrol.2017.01.011, 2017.

**Fig 2**: unit of precipitation is mm/day (LEFT) and mm/month (RIGHT).

**Response:** Done (L 198).

[Figure]

**Figure 2: Time series of meteorological data and rainwater isotopic values in 2019, monthly precipitation in 2019, and multi-year mean, respectively.**

**Fig 5**: I would rotate this figure 90 degrees, so you can more easily compare the figures with fig 3, 4, and 6.

**Response:** Agreed. The Figure 5 has been revised (L 227).

[Figure]

**Figure 5: Temporal dynamics of δD values in xylem water for 11- and 17-year-old apple trees. Sample collection started on day 1 before D₂O tracer solution (δD = 714000‰) injection and commenced until day 7 (N=6). Gray dashed lines represent the background value (mean δD values in xylem water on day 1 before labeling) and black dashed lines represent 2 SD above the background value.**

**L212:** "with relative higher reliance": I am not fully understand this sentence. Could you explain?

**Response:** It means that the contribution proportion of water from 140–320 cm soil layer both exceeded 48% in BYF stage for 11- and 17-year-old apple trees, which was higher than other stages. To clarify it, we have reorganized the sentence (L 254).

"The BYF stage produced more negative isotopic values in xylem water for 11- and 17-year-old apple trees (Fig. 7), which mainly utilized water from 140–320 cm soil layer (more than 48%)".

**Anonymous Referee #2**

**General comment**

The authors of this manuscript designed a tracer experiment based on stable isotopes of hydrogen and oxygen to identify the depth of soil water taken up by apple trees at three different stages (blossom and young fruit, fruit swelling and fruit maturation stage). The topic of this manuscript is timely and potentially interesting for the readers of Hydrology and Earth System Sciences. Overall, the manuscript is well written and structured, but I found that various and important methodological details (e.g., a detailed description of the extraction method used for soil and vegetation material, and the isotopic composition of the injected water) were not described in the manuscript. Furthermore, the authors should also have considered in the introduction and the discussion recent literature on isotopic fractionation and offset which presents the current technical limitations for the application of stable isotopes in ecohydrological studies. Finally, a section about the limitation of the methodological approach should be added before the conclusions.

**Response:** Thank you for your constructive and encouraging comments and giving us an opportunity to revise this paper. Corrections have been made based on the recommendations, with detailed response to each comment presented below.

(1) Methodological details have been added to the revised manuscript (L 119-121, 125-126 and 149-161).

[revised manuscript text omitted]

**Specific comments**

**Lines 44-45:** The authors should mention in the introduction that isotopic fractionation has been observed in various studies (e.g., Poca et al., 2019; Vargas et al., 2017; Barbeta et al., 2019), along the soil-root-stem-twig-leaf pathway. The factors affecting such fractionation/offset are still unclear, but they seem to be mainly related to the water extraction technique, and particularly to cryogenic vacuum distillation (e.g., Zhao et al., 2016; Barbeta et al., 2022).

**Response:** Agreed. Information on isotopic fractionation has been added to the

Introduction (L 50-54).

[revised manuscript text omitted]

**Lines 90-92:** What is the depth to the water table in the study area? Do the roots of the apple trees have access to shallow groundwater?

**Response:** The groundwater level in the study area is over 50 m deep on average, which cannot be reached by plant roots. We have amended the manuscript (L 97).

**Lines 92-93:** What is the average, minimum and maximum distance between the trees? Does the tree distance affect the root density and possibly the water uptake?

**Response:**

(1) The plant and row spacing for the two orchards was 4 × 4 m, with all trees planted at the same interval.

(2) The results of two-dimensional root distribution of apple tree roots from Huo et al. (2021) showed that apple trees in shallow soil layers had relatively high root density and wide horizontal root distribution. With soil depth and horizontal distance increasing, the root density of apple trees significantly decreased. The tree root density was low and horizontal distribution was narrow in deep soil layers. Therefore, tree distance had little effect on root density and water uptake.

We have made changes in the revised manuscript (L 101).

**References**

Huo, G., Gosme, M., Gao, X., Dupraz, C., Yang, J., and Zhao, X.: Dynamics of interspecific water relationship in vertical and horizontal dimensions under a dryland apple-*Brassica* intercropping Quantifying by experiments and the 3D Hi-sAFe model, Agr. Forest Meteorol., 310, 108620, 10.1016/j.agrformet.2021.108620, 2021.

**Table 1:** It is unclear whether height, trunk and crown size data represent an average or single values. The authors should add the minimum and maximum values for all characteristics, as well as the number of trees used in this study.

**Response:** The height, trunk diameter (TD) and crown size (CS) were mean values for 20 trees in each orchard. The maximum and minimum values for all characteristics are shown in Table S1.

**Table S1 The maximum and minimum values of the height (H), trunk diameter (TD) and crown size (CS) for apple trees in two orchards.**

| Stand age (a) | $H_{max}$ (cm) | $H_{min}$ (cm) | $TD_{max}$ (cm) | $TD_{min}$ (cm) | $CS_{max}$ (cm) | $CS_{min}$ (cm) |
|---|---|---|---|---|---|---|
| 11 | 403 | 320 | 12.8 | 11.5 | 430 × 400 | 360 × 320 |
| 17 | 450 | 355 | 15.2 | 13.8 | 475 × 420 | 390 × 340 |

**Lines 108-109:** The authors should report the isotopic composition of the water mixture used for injection/irrigation, as well as the total amount of irrigation water applied per each tree.

**Response:** The isotopic composition of the tracer solution and total amount of injected solution have been added to the revised manuscript (L 119-121).

"A long polyvinyl chloride pipe was inserted into the holes at the target depth before

injecting 300 mL tracer solution ($\delta D$ = 714,000‰, 30 mL 99.99% $D_2O$ plus 270 mL tap water) into each hole. The total amount of injected solution was 1,200 mL for each tree."

**Lines 109-111:** I do not understand the aim of this sentence. Based on it, I understand that the authors may have injected a water amount that could be too small to detect a soil water content variation and perhaps, also an isotopic variation.

**Response:**

(1) The results from Huo et al. (2020) showed that 300 mL of aqueous solution wet 400 $cm^3$ of the soil on the Loess Plateau, equivalent to a <1% change in SWC (within the measurement error), so the impact on soil hydrological processes was negligible.

(2) Unlike soil water content, the isotopic composition of soil water changed obviously after injection due to the high isotopic value of D (714,000‰) in the injected solution, which could reach to 150,000‰.

**Lines 112-113:** Details about the sample size should be reported in the text here or in an additional table. Furthermore, more details are needed to understand how the background value was computed (is it an average of how many samples?), and where these unlabeled trees are located compared to the trees used for the experiment.

**Response:** Agreed. Two xylem samples were collected for each tree, with a total sample size of six for each treatment in a single sampling. The background value was an average of six xylem samples from unlabeled trees. The text has been revised (L 125-127). The layout of unlabeled trees and labeled trees are shown in Figure S1.

[Figure]

**4 m**

**4 m**

**24 m**

**0.5m**

⊗ Experimented apple tree    ● Injection hole

Unlabelled tree     Injection at 1m depth     Injection at 2m depth     Injection at 3m depth     Injection at 4m depth

**Figure S1 The layout of apple trees in experiment plot.**

**Lines 132-133:** I am not familiar with this cryogenic vacuum distillation system, and I am not sure it is manufactured by Los Gatos Research. I recommend adding here a detailed description of the extraction method, as well as information about the extraction efficiency.

**Response:**

(1) The manufacturer of the cryogenic vacuum distillation system has been revised (L 149-151).

"A cryogenic vacuum distillation system (Li-2000; LICA United Technology Limited, Beijing, China) was used to extract water under a heating temperature of 95 ℃ and a pressure of 0.2 Pa, which has been applied in previous studies (Huo et al., 2020; Tao et al., 2021a; Wang et al., 2021b; Zhao e al., 2020)."

(2) A detailed description of the cryogenic vacuum distillation method and its

extraction efficiency has been added to the revised manuscript (L 149-153).

"A cryogenic vacuum distillation system (Li-2000; LICA United Technology Limited, Beijing, China) was used to extract water under a heating temperature of 95 ℃ and a pressure of 0.2 Pa, which has been applied in previous studies (Huo et al., 2020; Tao et al., 2021a; Wang et al., 2021b; Zhao e al., 2020). The extraction time of soil water and xylem water samples were 90 min and 120 min respectively. Samples were weighed before and after extraction and again after oven-drying for 24 h to calculate the extraction efficiency (Wang et al., 2021b), which should be not less than 98%."

**Figure 5:** The isotopic composition of the injected water should be plotted, and I suggest showing the background value also without the 2 SD. Furthermore, the sample size should be reported in the caption.

**Response:**

(1) Because the concentration of D in tracer solution ($\delta D$ = 714,000‰) is much higher than it in xylem water, it cannot be shown in the Figure 5. We have added the isotopic composition of the tracer solution to Figure 5's title (L 229).

(2) The background value without 2 SD has been added to Figure 5 (L 227).

(3) The sample size has been added to Figure 5's title (L 229).

[Figure]

**Figure 5: Temporal dynamics of $\delta D$ values in xylem water for 11- and 17-year-old apple trees. Sample collection started on day 1 before $D_2O$ tracer solution ($\delta D$ = 714000‰) injection and commenced until day 7 (N=6). Gray dashed lines represent the background value (mean $\delta D$ values in xylem water on day 1 before labeling) and black dashed lines represent 2 SD above the background value.**

**Figure 7:** In the plots for "FSW for 11yr" and "BYF for 17yr" there may be an isotopic offset for some xylem water samples; I suggest checking whether there is a significant deviation from the soil water isotopic line, by also considering the uncertainty due to the isotopic analyses. In these plots, the authors should add the isotopic composition of the background values (in soil and xylem waters) and of the

injected water. Equation of the LMWL and details about sample size and when the samples were collected should be added in the caption.

**Response:**

(1) We assessed the isotopic offset between xylem water and soil water using soil water line conditioned excess (SW-excess) proposed by Barbeta et al. (2019):

$$\text{SW-excess} = \delta D - a_s\delta^{18}O - b_s \tag{1}$$

where $a_s$ and $b_s$ are the slope and intercept of soil water line (SWL), respectively; $\delta D$ and $\delta^{18}O$ are the isotopic compositions of xylem water.

**Table S2 Mean (±SD, n=6) soil water excess (SW-excess, ‰) for 11- and 17-year-old apple trees.**

| Stand age (a) | Growth stage | | |
|---|---|---|---|
| | BYF | FSW | FTM |
| 11 | 2.09±0.83 | -6.80±3.56 | 4.78±1.85 |
| 17 | -2.75±075 | -2.57±0.29 | -0.80±1.72 |

The results showed that SW-excess was negative in FSW stage for 11yr and BYF stage for 17yr trees, which meant that xylem water plots below SWL in a $\delta D - \delta^{18}O$ diagram (i.e. D in xylem water is more depleted than SWL). Therefore, the $\delta D$ after subtracting the SW-excess and $\delta^{18}O$ values for xylem water were used as mixture data for the Bayesian isotope mixing model.

The text has been revised (L 162-168 and 174-177).

(2) The soil water line and xylem water line have been added to Figure 7. We did not show the isotopic composition of injected water because the soil water and xylem water in these plots were from unlabeled trees (L 264).

[Figure]

**Figure 7: $\delta$D and $\delta^{18}$O values in xylem water and different soil layers (0–40 cm, 40–140 cm, 140–240 cm, 240–320 cm) for 11- and 17-year-old apple trees (±SD). The GMWL and LMWL represents the global and local meteoric water lines. LMWL: $\delta$D = 7.1$\delta^{18}$O+2.1. The SWL and XWL represents the soil water line and xylem water line, respectively.**

(3) The LMWL Equation has been added (L 266). "LMWL: $\delta$D = 7.1$\delta^{18}$O+2.1".

Details on sample size and sample timing have been added to the "Materials and Methods" and Figure 2 (L137-140).

"Rainwater samples (N = 32) were collected using a combined device of polyethylene bottle and funnel during rainfall events between May and September. A plastic ball was placed on the funnel to prevent evaporation. The collected rainwater samples were immediately sealed into vials by parafilm and stored at 4°C for isotopic determination."

[Figure]

**Figure 2: Time series of meteorological data and rainwater isotopic values in 2019 and monthly precipitation in 2019 and multi-year mean, respectively.**

**Figure 9:** Please remove the regression line when there are only three samples used for the analysis.

**Response:** Done (L 283).

[Figure]

**Figure 9: Relationship between the contribution of water sources and soil water content in different soil layers of 11- and 17-year-old apple orchards.**

**Section 3.6:** I expected to read here the results concerning the application of the index S, but they are not reported. Therefore, I suggest adding here such results, or removing the description of S at Lines 145-148.

**Response:** Agreed. The description of S has been deleted in the revised manuscript (L 180-183).

**Section 4.3:** Please add in this section (or in a new one) the limitations of the experimental approach. I think the limitations of this work are mainly related to the assumptions of no isotopic fractionation and negligible spatial variability in the

isotopic composition of unlabeled and labeled trees, and to the extraction method (cryogenic vacuum distillation system).

**Response:** We have added a new section to the revised manuscript (L 389-405).

**"4.4 Uncertainty caused by isotopic offset"**

"In this study, isotopic offset between xylem and soil water was observed for both 11- and 17-year-old unlabeled apple trees (Fig.7 and Table S2). We used the isotopic composition of soil water to correct $\delta$D values of xylem water, ensuring they match those of soil water. Although we did not collect soil water isotope samples in the isotope labeling experiments, this may have little effect on determining the soil layer depths from which trees derive their water source due to the high $\delta$D values in the injected solution. It should be noted that isotopic spatial heterogeneity related to destructive sampling (xylem and soil water) could lead to an isotopic mismatch between xylem and soil water. Recently, isotopic offsets between plants and their potential water sources have been also found in various ecosystems, which may hinder the unambiguous identification of water sources and influence the accurate assessment of DLSW utilization (Barbeta et al., 2022; De La Casa et al., 2022; Zhao et al., 2016). Some studies argued that isotopic fractionation during root water uptake could be attributed to the existence of Casparian strips which can lead to isotope enrichment in root water and depletion in xylem water (Naseer et al., 2012; Vargas et al., 2017). Seeger and Weiler (2021) questioned whether xylem water was completely renewed by newly absorbed soil water, thus affecting isotopic offset. Furthermore, CVD may mask or exaggerate the isotopic offset, although it was the most common methodology (Chen et al., 2020; Orlowski et al., 2016a,b, 2018; Wen et al., 2022). When quantifying the water use strategies of plants, the isotopic measurement bias related to CVD should be considered. As a whole, there are various trends and causes of isotopic offset; further research about offset is urgently needed to better understand root water uptake processes."

**References**

[revised manuscript text omitted]

**Title of section 2.2:** I suggest changing it with "Sample collection".

**Response:** Done (L 109).

"**2.2** Sample collection"

**Title of section 2.2.2:** I suggest changing it with "Collection of soil and vegetation samples for isotopic analysis".

**Response:** Done (L 129).

"2.2.2 Collection of soil and vegetation samples for isotopic analysis"

**Figure 4:** Please add in the caption when the soil water content was determined (before, during or after the tracer injection).

**Response:** Done (L 213).

"Figure 4: Vertical distribution of soil water content (SWC) before the tracer injection in 11-year-old (A) and 17-year-old (B) apple orchards. Values are means ± SD (N=3)."

**Figure 8:** Please remove from the caption "Seasonal patterns of" because the results refer only to three specific tracer injections.

**Response:** Done (L 270).

"**Figure 8:** The contribution of four potential water sources to xylem water in 11-year-old (A) and 17-year-old (B) apple trees. Error bars indicate standard errors of the means (N=3). Asterisks represent significant differences between growing stages (*, $P < 0.05$; **, $P < 0.01$; ***, $P < 0.001$)."

---

## Author Response (AR2)

HESS-2022-142

Title: The natural abundance of stable water isotopes method may overestimate

deep-layer soil water use by trees

Author(s): Shaofei Wang et al.

MS type: Research article

Iteration: Minor revision

**Comments from handling editor:**

Dear authors,

The Referee acknowledges the substantial improvement of the revised manuscript, but still quite a few minor issues and technical points need to be adjusted.

Best regards

Roberto Greco

**Response:** Thank you very much for providing the opportunity to revise our manuscript entitled "The natural abundance of stable water isotopes method may overestimate deep-layer soil water use by trees" (hess-2022-142). The reviewer's comments are constructive and helpful. We have gone through the comments carefully and made revisions. Based on the instructions in your letter, we have uploaded the revised manuscript with revisions being marked. The point-by-point responses to the reviewer's comments are presented as follows.

**Comments from anonymous referee #2**

Thank you for considering my previous comments. I think you have significantly improved the manuscript. However, I still have some minor comments and technical corrections. Lines (L) refer to the revised manuscript with no track changes.

**Response:** Thanks a lot for encouraging and constructive comments on our

manuscript. We have carefully revised the manuscript according the comments. Please see the details as follows.

**Minor comments and technical corrections**

L86: Unclear b) objective. I suggest replacing 'exploit' with another verb.

**Response:** We agree. It has been edited in the text (L 87).

"(b) ascertain the water use strategy response of apple trees to variations in the growing season and stand age"

L87: Please specify what the 'combined method' is.

**Response:** We agree. It has been edited in the text (L 88-90).

"(c) elucidate the difference between the combined method (combining isotopic labeling in deep soils with natural stable isotope signatures) and the natural abundance of stable water isotopes method.

Title of section 2.2.3: Please change it to 'Sampling of rain water'.

**Response:** We agree. It has been edited in the text (L 134).

"2.2.3 Sampling of rain water"

L135: Please replace 'determination' with 'analysis'.

**Response:** We agree. It has been edited in the text (L 137).

"The collected rain water samples were immediately sealed into vials by parafilm and stored at 4°C for isotopic analysis."

L148: I guess the authors meant that only samples with an extraction efficiency equal

or higher than 98% were considered for isotopic and data analysis. Please rephrase the sentence, by replacing 'should be' with another verb or tense.

**Response:** We agree. The sentence has been rewritten (L 150-151).

"Samples with an extraction efficiency less than 98% were discarded."

L208-209: Please revise the sentence because xylem samples for FSW for 11-year-old apple trees, collected after 7 days, have $\delta^2H$ values that are much more negative than the background (Fig. 5).

**Response:** We agree. The text has been revised (L 210-212).

"The maximum concentration of D in xylem for both apple orchards occurred on day 3 after labeling, then decreased rapidly, and was lower than background values on day 7."

Title of section 4.4: Please change it to 'Limitations due to the extraction method'.

**Response:** We agree. It has been edited in the text (L 368).

"4.4 Limitations due to the extraction method"

L370-371: It is unclear how spatial heterogeneity is related to the destructive sampling. Please rephrase the sentence.

**Response:** Agreed. We have rewritten this sentence (L 373-375).

"It should be noted that isotopic spatial heterogeneity of xylem water induced by sampling position and time (Nehemy et al., 2022) and soil water induced by uneven distribution of throughfall and preferential flow (Xiang et al., 2019; Yang and Fu, 2017) could lead to an isotopic offset."

**Reference**

Nehemy, M. F., Benettin, P., Allen, S. T., Steppe, K., Rinaldo, A., Lehmann, M. M., McDonnell, J. J.: Phloem water isotopically different to xylem water: Potential causes and implications for ecohydrological tracing, Ecohydrology, 15(3), e2417, DOI:10.1002/eco.2417, 2022.

Xiang, W., Si, B. C., Biswas, A., and Li, Z.: Quantifying dual recharge mechanisms in deep unsaturated zone of Chinese Loess Plateau using stable isotopes, Geoderma, 337, 773-781, 10.1016/j.geoderma.2018.10.006, 2019.

Yang, Y. and Fu, B.: Soil water migration in the unsaturated zone of semiarid region in China from isotope evidence, Hydrol. Earth Syst. Sci., 21(3), 1757-1767, DOI:10.5194/hess-21-1757-2017, 2017.